



# Global distribution of wastewater treatment plants and their released effluents into rivers and streams

Heloisa Ehalt Macedo[1], Bernhard Lehner[1], Jim Nicell[2], Günther Grill[1], Jing Li[2], Antonio Limtong[1], Ranish Shakya[1]

[1]Department of Geography, McGill University, Montreal QC H3A 0B9, Canada
[2]Department of Civil Engineering, McGill University, Montreal QC H3A 2K7, Canada

*Correspondence to*: Heloisa Ehalt Macedo (heloisa.ehaltmacedo@mail.mcgill.ca) or Bernhard Lehner (bernhard.lehner@mcgill.ca)

**Abstract.** The main objective of wastewater treatment plants (WWTPs) is to remove contaminants such as pathogens,
nutrients, and organic and other pollutants from wastewaters using physical, biological and/or chemical processes prior to discharge into receiving waterbodies. However, since WWTPs cannot remove all contaminants, they inevitably represent concentrated point sources of residual contaminant loads into surface waters. To understand the severity and extent of the impact of wastewater discharges from such facilities into rivers and lakes, as well as to identify opportunities of improved management, detailed information about WWTPs is required, including (1) their explicit geospatial locations to identify the
waterbodies affected; and (2) individual plant characteristics such as population served, flow rate of effluents, and level of treatment of processed wastewaters. These characteristics are especially important for contaminant fate models that are designed to assess the distribution of substances that are not typically included in environmental monitoring programs, such as contaminants of emerging concern. Although there are several regional datasets that provide information on WWTP locations and characteristics, data are still lacking at a global scale, especially in developing countries. Here we introduce
HydroWASTE, a location-explicit global database of 58,502 WWTPs and their characteristics. This database was developed by combining national and regional datasets with auxiliary information to derive or complete missing WWTP characteristics, including the amount of people served. A high-resolution river network with streamflow estimates was used to georeference WWTP outfall locations and calculate each plant's dilution factor (i.e., the ratio of the natural discharge of the receiving waterbody to the WWTP effluent discharge). The utility of this information was demonstrated in an assessment of the
distribution of wastewaters at a global scale. Results show that 1.2 million kilometers of the global river network receive wastewater input from upstream WWTPs, of which more than 90,000 km are downstream of WWTPs that offer only primary treatment. Wastewater ratios originating from WWTPs exceed 10% in over 72,000 km of rivers, mostly in areas of high population densities in Europe, USA, China, India, and South Africa. In addition, 2,533 plants show a dilution factor of less than 10, which represents a common threshold for environmental concern.



## 1 Introduction

In all inhabited regions of the world, the water quality of rivers, lakes and ultimately the ocean depends on how wastewaters produced from human activities in upstream areas, especially those that are densely populated, are processed and disposed. Globally produced domestic and municipal wastewater is estimated to amount to 360 km$^3$ year$^{-1}$, of which 41 km$^3$ year$^{-1}$ (11.4%) is treated in wastewater treatment plants (WWTPs) and then re-used, 149 km$^3$ year$^{-1}$ (41.4%) is treated in WWTPs and then discharged, and 170 km$^3$ year$^{-1}$ (47.2%) is not treated in WWTPs but released directly to the environment (Jones et al., 2021). According to recent assessments, approximately 3.1 billion people worldwide had access to sewage systems connected to WWTPs in 2017 (WHO & UNICEF, 2017).

Although the overall goal of WWTPs is to reduce the load of pollutants that reach downstream waterbodies, most WWTPs are only designed to remove organic matter and macro pollutants. Thus one of the biggest issues related to global wastewater treatment is the efficiency of removal of specific contaminants, particularly those related to new products or chemicals that are released without appropriate regulatory oversight and with uncertain or unknown effects on the environment and human health (WHO & UN Habitat, 2018). These "emerging contaminants" (e.g., pharmaceutically active compounds, microplastics, ingredients in household and personal care products) are not commonly monitored and most WWTPs are not designed to remove them either fully or partially before releasing effluents to nearby waterbodies. As a result, wastewaters are being collected from municipal sources, transported to a location where they may or may not be treated, and then released into the environment, thereby causing the WWTP to serve as a concentrated point source of contamination of receiving waterbodies (Daughton and Ternes, 1999; Musolff et al., 2008). Once the contamination enters the river network it continues to flow downstream, potentially accumulating with other contaminants from multiple sources along the way, to sometimes deleterious effects.

Studies have demonstrated that the fraction of wastewater is directly proportional to effects on biodiversity and ecosystems in rivers downstream of effluent discharge (Munz et al., 2017; Neale et al., 2017; Bunzel et al., 2013). Therefore, the dilution factor (i.e., the ratio between the natural discharge of the receiving waterbody and the WWTP effluent discharge) is one of the major determinants of ecological risks originating from WWTPs (Link et al., 2017). Dilution factors have been used to predict potential exposure to down-the-drain chemicals from population density (Keller et al., 2014), which at a regional level can help prevent negative effects and determine hotspots of contamination. However, to identify which particular WWTPs should be targeted for the implementation of more stringent treatment standards and/or be upgraded through the deployment of advanced treatment technologies, it is necessary to first determine where treated effluents are being discharged in order to pinpoint which individual waterbodies downstream are potentially affected by their wastewaters. For example, Rice and Westerhoff (2015) analyzed the effects of WWTP effluent locations upstream of drinking water treatment plants, and Vigiak et al. (2020) estimated the domestic waste emissions to European waters from WWTPs.

Therefore, and for regulatory purposes, national and regional governments, non-governmental organizations, and commercial data providers gather information about the exact geospatial location of WWTPs and their attributes such as population served,



wastewater discharged and level of treatment. Datasets on WWTPs are available at different scales, with different attributes, and highly variable frequencies of updates. Datasets from Europe (EEA, 2017) and from the United States (US EPA, 2016)

contain information on the location and characteristics of WWTPs (e.g., generated load, treatment capacity, level of treatment) and are easily accessed and readily available for download. In contrast, many countries, such as those throughout most of South America, Africa, Eastern Europe, and Asia do not provide readily accessible information about their wastewater sector. The main sources of global wastewater information, available at the country-level, are the Joint Monitoring Program (JMP-WASH) (WHO & UNICEF, 2017) and the Global Information System on Water and Agriculture (AQUASTAT) (FAO, 2016).

JMP-WASH provides regular global reports on drinking water and sanitation coverage for tracking progress toward the Sustainable Development Goal for clean water and sanitation (SDG 6) that has been defined by the United Nations (Herrera, 2019). AQUASTAT provides data on water resources and water use, with emphasis on agricultural water management.

Also at the global scale, Jones et al. (2021) recently produced a modeled, spatially disaggregated map of the amounts of wastewater production, collection, treatment and re-use. Besides estimating previously unavailable country-level wastewater

statistics, the authors downscaled the country-level data to a 5 arc-minute resolution grid using return-flow data from the global hydrological model PCR-GLOBWB 2. Nonetheless, the new dataset does not include the exact point location of treatment plants, the location of their individual discharge into the stream network, nor the level of treatment of processed wastewaters. Some web interface platforms such as Wikimapia and Open Street Map also provide WWTP locations as point coordinates. Both platforms are built based on contributions from users around the world and are free for public use. The information is

global and constantly updated. However, user input may not be equally distributed in space, thus some regions are incomplete. Furthermore, the locations are not necessarily verified and information other than point coordinates are often missing. Currently no comprehensive global database of geolocated WWTPs exists.

One of the main applications of WWTP datasets with spatially explicit locations is in water quality modelling, representing point sources of contaminants discharged into the river and lake system. In Europe, the Urban Waste Water Treatment

Directive (UWWTD) (EEA, 2017) has been used as input data in water quality models for pharmaceuticals and nanoparticles, such as ePiE (Oldenkamp et al., 2018) and GWAVA (Dumont et al., 2015). In the United States, WWTP information from the Clean Watersheds Needs Survey (CWNS) (US EPA, 2016) has been incorporated in the models PhATE (Anderson et al., 2004) and iSTREEM (Kapo et al., 2016) to assess the concentration of pharmaceuticals and other chemicals in river systems. In Canada and China, WWTP information from government sectors was used as an input for the contaminant fate module of

the river routing model HydroROUT (Grill et al., 2016; Grill et al., 2018).

Whereas these existing contaminant fate models operate from local to regions scales, i.e. from catchments to continents, water pollution is of global concern. Robust estimates of current and future changes in water quality are needed to support global environmental and health risk decision making and to sustainably manage water resources to ensure clean and accessible water for all, as required by SDG 6 (Van Vliet et al., 2019; Tang et al., 2019; Strokal et al., 2019). To achieve this goal, global water

quality assessments must be spatially consistent and comparable to be able to identify locations of high contaminant concentration (i.e., "hotspots") and trends in water pollution over time and across large regions. Global water quality models





can also account for large-scale drivers that might not be captured by small-scale models (Tang et al., 2019). One of the main challenges for global water quality modelling is the lack of spatial consistency in datasets for model inputs, especially in regions where data are insufficient for a detailed assessment (Strokal et al., 2019; Tang et al., 2019; Kroeze et al., 2016). Due to the limited information on global wastewater, all published global water quality models until now (e.g. GLOBAL-FATE, Global NEWS, WorldQual, GlowPa, IMAGE-GNM) quantify the load of wastewater into the river system using population density and national sanitation statistics as proxies (e.g., Font et al., 2019; Strokal et al., 2019; Mayorga et al., 2010; Van Drecht et al., 2009; Williams et al., 2012; Beusen et al., 2015; Hofstra et al., 2013). More specifically, calculations are based on the fractions of population connected to sewage systems per country.

To address this important shortcoming, the objective of the present study is to develop a novel global database of WWTPs as a means for estimating the distribution of wastewaters in the global river network at high spatial resolution. The database, termed HydroWASTE, includes the explicit geospatial locations of WWTPs, their linkages with the global river and lake network, as well as their main characteristics.

## 2 Data and methods

### 2.1 Development of HydroWASTE

To create HydroWASTE, three main steps were undertaken, as shown in Fig. 1: (1) the combination of national and regional datasets, including the correction of errors using the WWTP point locations and attributes available; (2) the georeferencing of WWTPs to a global river network, in order to connect the facilities to their receiving waterbodies; and (3) the estimation of missing attributes for each WWTP, including population served, wastewater discharge and level of treatment, using geospatial methods and auxiliary datasets such as modeled river discharge estimates, gridded global population numbers, gross national income per capita, and country-level statistics on sanitation.

The design of HydroWASTE was tailored for its potential application in water quality modelling. The main attributes that are typically required to simulate the wastewater component in water quality models include (Grill et al., 2016; Grill et al., 2018): (1) the WWTP's location (point coordinates); (2) the estimated effluent outfall location (linkage between WWTP and river network); (3) the number of people served by the WWTP; (4) the amount of wastewater discharged; and (5) the level of treatment offered by the WWTP (primary, secondary, or advanced). The WWTP location is a necessary requirement for any spatially explicit assessment that is based on point sources of effluents discharged through WWTPs. Beyond knowing the actual location of the plant, it is also important to provide the approximate effluent outfall location into the local river network, which can differ substantially from the WWTP location. The number of people served by WWTPs is required to estimate contaminant loads that reach the facility, while the wastewater discharge and the corresponding level of treatment provide the basis for calculating the contaminant loads that are discharged by the facility into receiving waterbodies. If no data concerning the population served are available, wastewater discharge can be used in lieu of this, provided that a reasonable conversion factor between the two can be estimated (see Section 2.1.4 below). Some of these attributes can be directly compiled from





national or regional WWTP datasets, after applying the necessary unit conversions and quality checks. Other attributes must
be estimated based on geographical and statistical methods.

### 2.1.1 Cleaning, filtering, and combing WWTP national datasets

After intensive literature and online searches, several national (or multi-national/regional in the case of Europe) WWTP
datasets were identified that provide the geographic location of WWTPs, as well as a varying list of additional attribute
information such as population served, amounts of effluents discharged, and level of treatment (Table 1). In cases of multiple
datasets being available for the same country, such as in the case of the USA or for individual European countries, the most
comprehensive or most consistent dataset was chosen rather than merging all available data in order to avoid issues of duplicate
records. In most cases, datasets were retrieved from pertinent government agencies through publicly accessible website
platforms or personal communication. The quality, completeness, and consistency of the datasets strongly vary among the
different sources and nations. For all countries where no national data repositories were available, WWTP point locations
(without further attribute information) were added from the open-source web platform of Open Street Map (OSM;
https://www.openstreetmap.org/).

The selected datasets listed in Table 1 use different attribute nomenclatures and reporting units. For example, in the European
dataset, the population size is reported in 'population equivalents'; that is, it assumes one person produces 54 grams of
dissolved organic pollutants, expressed as biological oxygen demand (BOD) per 24 hours. Therefore, it accounts not only for
permanent residents of the surrounding area, but also for ambient populations, i.e. for differences between daytime and night-
time populations, including tourists (Nakada et al., 2017). The term 'population served', as used in most national datasets,
generally refers to the population physically connected to the particular WWTP, thus paying fees for the service (Daughton,
2012).

Filtering was necessary for some datasets that include additional records not regarding WWTPs, especially for the most
comprehensive datasets for the USA and Europe. These datasets include records of decentralized wastewater treatment
systems, stormwater facilities, and other wastewater collection systems that are not connected to a WWTP. Some datasets
include records with geographic coordinates outside the expected national or regional boundaries, which were assumed to be
errors and removed from HydroWASTE. More details about each dataset can be found in the Supplementary Information,
section S1.

### 2.1.2 Auxiliary datasets

#### *a) River network attributes*

To assign the estimated effluent outfall location of each WWTP, various raster and vector layers representing the river network
and catchment boundaries were obtained from the global HydroSHEDS database (Lehner et al., 2008), which was derived
from digital elevation data provided by NASA's Shuttle Radar Topography Mission (SRTM) at 90 m (3 arc-second) resolution.



For our study, we used a standardized derivative of this database, termed HydroATLAS (Linke et al., 2019), that offers sub-basin delineations at 12 hierarchical levels of increasingly finer subdivisions. We applied the smallest sub-basin breakdown of level 12 which provides 1,034,083 sub-basins globally with an average area of 130.6 km$^2$ (std. dev. 146.9 km$^2$). HydroATLAS also offers a preprocessed river network, including discharge information, that was extracted at 500 m (15 arc-second) grid cell resolution and represents all rivers and streams where the long-term (i.e., 1971-2000) average discharge exceeds 100 L s$^-$

$^1$ or the upstream catchment area exceeds 10 km$^2$, or both. Natural river discharge estimates were provided by the global hydrological model WaterGAP (Müller Schmied et al., 2014) version 2.2 as of 2014, which were downscaled from their original resolution of 0.5º grid cells to the HydroSHEDS resolution of 500 m using geostatistical techniques (Lehner and Grill, 2013).

### b) Country-level wastewater statistics

To infer missing attributes in the WWTP records, global datasets with information on wastewater at a country-level were used. Treated wastewater discharge at the country-level was provided by Jones et al. (2021) who collected and standardized data from online sources, especially the Food and Agriculture Organization's (FAO) Global Information System on Water and Agriculture (AQUASTAT), the Global Water Intelligence (GWI), Eurostat, and the United Nations Statistics Division (UNSD). The study provides data for the year 2015, and, where data were unavailable, the authors used multiple linear

regressions to estimate the values.

The World Health Organization and the United Nations Children's Fund (WHO/UNICEF) Joint Monitoring Program (JMP) for Water Supply, Sanitation and Hygiene (WASH) is responsible for monitoring the SDG target related to WASH (WHO & UNICEF, 2017). For this study, we acquired sanitation data for each country for the year 2017. The information selected is termed '*Proportion of population using improved sanitation facilities (sewer connections).*'

### c) Population grid

Global gridded population distributions of the year 2015 from the WorldPop dataset (WorldPop & CIESIN, 2018) were disaggregated from their original spatial resolution of 1 km to the same resolution (500 m) as the applied HydroATLAS data to allow for spatially consistent calculations. WorldPop was produced using a combination of census, geospatial, and remotely-sensed data in a spatial modelling framework (Tatem, 2017).

### d) Gross national income (GNI) per capita

The World Bank divides economies into four income groups (i.e., low, lower-middle, upper-middle, and high) based on Gross National Income (GNI) per capita (in U.S. dollars), calculated using the World Bank Atlas method (World Bank, 2019). This indicator refers not only to the economy, but also correlates with other non-monetary measures of quality of life. Here, the GNI of 2019 was used to classify countries based on their capacity to deploy different levels of wastewater treatment.





### 2.1.3 Georeferencing WWTP outfall locations to the global river network

A requirement for any spatially explicit water quality assessment that includes WWTPs is to know the approximate location at which each plant's effluents are discharged into a waterbody; i.e. typically a river, a lake, or the ocean. In reality, the location of the effluent discharge into the environment may be distinct from the WWTPs actual location, influenced by several local factors not easily obtainable and applicable at a global scale, such as environmental policies, political and social conventions, ecosystem characteristics, land use, and local conditions such as the presence of interfering pipelines and canals. As such, the reported WWTP locations used in this study are not warranted to represent their actual outfall locations, nor to intersect with the natural river network. In addition, due to inherent quality limitations of the global HydroATLAS river network, which was derived from a digital elevation model, and the applied spatial resolution of 500 m, the river location does not always correspond to reality, especially for small streams.

Given these uncertainties, we developed a rule-based procedure within a Geographic Information System (GIS) to estimate a representative point of connection between each WWTP and the river network (referred to herein as the estimated outfall location) using the following ruleset: (1) the outfall location should be within a predefined radius from the given WWTP point location; (2) only locations with average natural stream flows exceeding 100 L s$^{-1}$ or with an upstream catchment area exceeding 10 km$^2$ are considered as possible outfall locations to avoid allocation to very small streams; (3) if multiple options are available, priority should be given to larger rivers under the assumption that effluents are generally directed towards larger rivers to increase dilution); and (4) the location should be within the same sub-basin as the WWTP itself to avoid mis-allocation to close rivers across a watershed divide. By design, this ruleset assigns the outfall location to be downstream of the WWTP location (towards larger rivers), yet within a maximum radius, and this downstream allocation will generally reduce cases where effluents are (possibly erroneously) assigned to very small streams which could cause excessive estimates of wastewater concentrations in follow-up water quality assessments. We thus consider the described procedure to deliver a best-guess association within the given river network with an intended bias to deliver conservative results in terms of environmental risk studies. It is also important to note that the estimated outfall locations should not be interpreted as true and precise geographic locations.

The predefined radius wherein the estimated outfall location can be assigned to a river was set at 10 km. This choice was based on a statistical determination process using a subset of WWTPs and remote sensing imagery for manual verification (see Supplementary Information, section S2.3). If the closest location of connection to a river is further than 10 km, then the estimated outfall of the WWTP was georeferenced to that location, independent of distance, provided that all other rules still apply. In cases where the WWTP location is close to the sub-basin outlet, limiting the estimated outfall location to less than 10 km away from the WWTP location, the outfall location was additionally moved one grid cell (~500 m) further downstream; that is, into the next sub-basin and thus to a larger river, while keeping it close to the original WWTP location and in the same overarching basin (Fig. 2).



### 2.1.4 Estimation of missing attributes

As a prerequisite for many applications, such as the development of a global contaminant fate model, the characteristics of WWTPs should be consistent throughout the database. Based on previous studies of contaminant fate in rivers (Grill et al.,
2016; Grill et al., 2018; Strokal et al., 2019), the three most important attributes required to produce realistic contaminant load estimates are: (1) the number of people served; (2) total wastewater discharged by the plant; and (3) the level of treatment (i.e., primary, secondary, or advanced).

The availability of these three attributes in the original source data is highly variable between countries (Table 1). For instance, while data for the USA, New Zealand, Brazil, and China provide information on all three attributes, all other regions lack at
least one of them, including Europe, India, Canada and Mexico with two attributes and large parts of Africa, South America, Asia, and Australia only offering the WWTP location. For all incomplete data records, we thus inferred the missing attributes based on auxiliary information related to wastewater, such as reported country-level statistics on water use, sanitation, and economy, as well as population distributions.

Table 2 provides an overview of the extent of missing data and the auxiliary data that were used to fill the gaps. Processing
steps are explained in more detail below. Note that the order in which the missing data were estimated is predetermined: we first completed the records of population served as the results then informed the estimation of wastewater discharge and level of treatment.

### *a) Population served*

For WWTP records that did not include information on the population served by the plant, we estimated this attribute using
up to three different approaches (A1, A2 and A3; see Supplementary Information, section S3 for more information), depending on data availability, based on the following assumptions: (A1) the population served is directly related to the wastewater discharge of the WWTP; (A2) the population served should reside within relatively close proximity to the WWTP; and (A3) the treatment capacity of the WWTP cannot overload the receiving river's capacity for dilution. The latter assumption is based on the fact that governments typically regulate WWTP effluents to remain within specified dilution limits to mitigate adverse
effects of pollution on aquatic ecosystems downstream (Link et al., 2017; Munz et al., 2017; Neale et al., 2017). Once the different population values were estimated, the minimum value was selected to represent the limit of the WWTP's capacity in terms of population served. We chose the minimum to avoid excessive estimates of population served in subsequent water quality assessments.

For the first approach (A1) we estimated the number of people served, $P_{est}$, using the ratio between the plant's wastewater
discharge, $W_{rep}$ (as reported in the WWTP national dataset) and country-level statistics of treated wastewater per-capita, $U$ (as reported by Jones et al., 2021):

$$P_{est} = \frac{W_{rep} (L\ day^{-1})}{U\ (L\ day^{-1}\ capita^{-1})} \tag{1}$$





We tested the validity of the relationship described by Eq. (1) using countries with complete data availability (see Supplementary Information, section S3.1 for details) which confirmed a strong overall correlation ($R^2 = 0.80$; n = 28,497). If

the total treated wastewater for a certain country was recorded as 0 in the reference dataset, $U$ was substituted by the average treated wastewater per capita for the countries in the same economic group based on their GNI (World Bank, 2019).

For the second approach (A2) the method to estimate the maximum population served depended again on whether the WWTP record contained information on wastewater discharge or not. If no wastewater discharge attribute was included, the maximum population served was estimated as the total population surrounding the WWTP within a radius of 11 km, using WorldPop

population counts. This radius size was determined based on the outcome of a sensitivity analysis (see Supplementary Information, section S3.2). In the geospatial analysis, we ensured that each person in a region was served by only one plant, thereby avoiding double counting. In contrast, if a wastewater discharge attribute was available, the total population surrounding each WWTP was computed within a radius of variable size, based on the initial value of population served as calculated using approach A1. All WWTP records were grouped into four size categories of population served: <50,000 people;

50,000–100,000 people; 100,000–500,000 people; and ≥500,000 people. The radius assigned for each group was 5, 10, 20, and 30 km, respectively. This radius assignment was based on tests using the national dataset of India (see Supplementary Information, section S3.3).

For the third approach (A3), we used the dilution factor, $DF$, as defined by Eq. (2) to determine the limit of the WWTP's wastewater discharge, $W$, into the receiving river's natural discharge, $Q$, at the estimated outfall location (see section 2.1.3

above). $Q$ is provided by the HydroATLAS dataset (see section 2.1.2 above).

$$DF = \frac{Q + W (L\ day^{-1})}{W\ (L\ day^{-1})} \tag{2}$$

The minimum $DF$ recommended by the European Medicines Agency for environmental risk assessments of medicinal products for human use is 10 (EMA, 2006). However, this can sometimes differ in reality. Rice and Westerhoff (2017) found a wastewater ratio higher than 50% for over 900 streams receiving wastewater in the USA; i.e., representing a DF equal or lower

than 3. For the development of HydroWASTE, we therefore applied a minimum $DF$ of 5, i.e. WWTPs can be assigned maximum populations that would lead to effluent loads exceeding the EMA recommendation, yet within the range of ratios that are observed in reality. For WWTPs that have estimated outfall locations within 50 km of the ocean or a large lake (defined as those with a surface area larger than 500 km² in the global HydroLAKES dataset, Messager et al., 2016), we assume that environmental regulations are less restrictive since there is a large waterbody nearby that could greatly dilute the effluent. For

this reason, A3 is not applied for these WWTPs. The maximum population served, $P_{max}$, that the river could support was then calculated by solving Eq. (2) for $W$ (using $DF_{min} = 5$) and inserting it into Eq. (1), resulting in:

$$P_{max} = \frac{Q\ (L\ day^{-1})}{U\ (L\ day^{-1} capita^{-1})(DF_{min})} \tag{3}$$





In cases where the wastewater discharge is not reported (Table 2), only approaches A2 and A3 were used, which causes a higher level of uncertainty in these cases.

Finally, the minimum value among approaches A1, A2, or A3 was selected as the WWTPs estimate of population served. A correction was applied if the sum of the estimated population served by WWTPs in a country, $P_{tot}$, exceeded the total national population connected to sewers, $P_{stat}$, as reported by the JMP-WASH database. In this case, the estimated population served by each WWTP was multiplied by a reduction factor ($F$) to ensure that the total population served per country would not surpass national statistics:

$$F = \frac{P_{stat}}{P_{tot}} \qquad (4)$$

This correction was not applied for any country that reported population served in its national WWTP dataset.

### b) Wastewater discharge

We estimated wastewater discharge for all WWTP records that did not report on this attribute. Since a WWTP's wastewater discharge is directly related to the population served, Eq. (1) was modified to estimate the wastewater discharge ($W_{est}$) from 295 the reported or estimated population served ($P$) of the WWTP record:

$$W_{est}(L\ day^{-1}) = P * U\ (L\ day^{-1}\ capita^{-1}) \qquad (5)$$

### c) Level of treatment

The level of treatment of each WWTP was estimated based on the GNI per capita per annum categorization as defined by the World Bank for all countries, generally reflecting the observation that high-income countries have a higher probability of 300 advanced wastewater treatment than low-income countries. The applied relationships between income, population served, and level of treatment were determined based on national datasets that reported the level of treatment (see Supplementary Information, section S3.4 for details). As a result, for countries in the high-income group (USD 12,536 or more per annum), if the population served by the WWTP exceeds 3,000 (i.e., in predominantly urban settings), the level of treatment was set as advanced; otherwise, secondary treatment was assumed. For middle-income countries (USD 1,036 to USD 12,535 per annum), 305 the level of treatment was set as secondary. We did not find any WWTP regional datasets for countries from the low-income group (USD 1,035 or less per annum). We assumed that the level of treatment is the most basic, i.e. primary, in these countries, which may lead to some underestimations of their actual treatment potential.

### 2.2 Application of HydroWASTE to estimate dilution factors and wastewater ratios in global rivers

The dilution factor was calculated for all WWTP records in HydroWASTE using Eq. (2) and the natural river discharge ($Q$; 310 as reported in the HydroATLAS database) at the estimated outfall location. For WWTPs where the outfall location coincides with a lake from the HydroLAKES dataset (Messager et al., 2016), $DF$ was calculated based on the natural discharge at the





outflow of the lake to the river network. Since there is no meaningful value for direct discharge into the ocean or a large lake (i.e., lakes with a surface area larger than 500 km$^2$), the *DF* for WWTPs where the estimated outfall location is within 10 km of the ocean or a large lake is assumed to be infinite.

Finally, the distribution of wastewaters in the global river network was assessed by calculating the ratio of wastewater to natural discharge in every river reach. For this, the wastewater quantities discharged from all WWTPs were routed and accumulated downstream, from the estimated effluent outfall locations to the ocean, and divided by the long-term natural discharge as provided for all river reaches in the HydroATLAS database (see 2.1.2 above). The WWTPs reported as "Closed", "Decommissioned" or Non-Operational" were included in this analysis for their potential as source of residues in river

sediments from former discharge (Thiebault et al., 2021). This process was performed using the river routing model HydroROUT (Lehner and Grill, 2013).

## 3 Results

### 3.1 HydroWASTE: a global WWTP database

HydroWASTE contains a total of 58,502 WWTPs, each including a reported or estimated attribute of population served,

wastewater discharge, and level of treatment. From these, 58,278 records were successfully georeferenced to the global river network of HydroATLAS. The remaining 224 WWTPs were not linked to the river network as they were located on small islands or in small coastal basins and are thus assumed to discharge directly to the ocean. The average distance between the WWTP location in the source data and its estimated effluent outfall location is 6.5 ± 3.1 km with a maximum distance of 21.8 km.

Figure 3 presents the spatial distribution of WWTPs in HydroWASTE. Europe and the USA show the highest densities of WWTPs, whereas China and India have somewhat lower densities but much larger facilities (i.e., more population served, see Table 3). Figure 3 also shows the comprehensiveness of the reported attributes of each regional dataset and an evaluation of HydroWASTE's population served against the JMP-WASH database (WHO & UNICEF, 2017). Since we limited our estimated values of population served so that they did not surpass the country-level records, most countries that show a large

error correspond to underestimations. Exceptions occur in many European countries; here, population served was calculated from reported values of 'population equivalents', which includes not only permanent residents but also ambient population and, thus, can exceed the reported national population values in the JMP-WASH database.

Table 3 provides an overview of the 20 countries with the largest numbers of population served by WWTPs in HydroWASTE. These countries contribute around 83% of the total global treated wastewater (Jones et al., 2021). Table 3 also includes the

attributes reported by JMP-WASH (WHO & UNICEF, 2017) and Jones et al. (2021) for each country for comparison. For population served, the results confirm that HydroWASTE tends to overestimate values for European countries compared to JMP-WASH data, whereas for other countries it tends to underestimate them (due to incomplete records), leading to an overall global underestimation of 22%. However, over- or underestimated population served does not directly translate to equally





over- or underestimated values of wastewater discharge. In fact, total global wastewater discharge from HydroWASTE

underestimates those reported by Jones et al. (2021) by only 2%. USA is the country with the best accordance regarding both attributes analysed, reflecting a presumed high level of data completeness and quality in HydroWASTE.

In terms of missing attribute information that was not reported but was instead complemented using statistical methods, we assigned 39% of the total population served and 33% of the total wastewater discharge in HydroWASTE through statistical estimates (Table 4).

In order to evaluate the robustness of the methods applied to estimate population served and wastewater discharge for records with missing information, we used a subset of 28,497 WWTPs in HydroWASTE that have reported values of both attributes (see Supplementary Information, section S3.1 and Table S1 for details on these data). We applied the same methods as for the completion of missing attributes to additionally create an estimated value of both reported attributes in this WWTP subset. Figure 4 shows the comparison between the reported and the estimated values. For population served, 97.6% of the estimated

values were within one order of magnitude of reported values, while for wastewater discharge 99.1% remained within one order of magnitude.

The method to predict the level of treatment for WWTPs that lacked this attribute was evaluated by applying it to all WWTPs with reported levels of treatment ($n = 47,315$). Overall, our model could correctly predict the level of treatment for 70% of plants (Table 5). The 'primary' treatment level could not be validated as this treatment level was predicted only for low-income

countries, yet no reported data were available for this income category to compare against.

### 3.2 Global dilution factors

The dilution factors ($DFs$) were calculated for every WWTP record using Eq. (2), except for WWTPs that were assumed to discharge into large lakes or the ocean ($n = 10,445$) for which we assigned an infinite $DF$ (see section 2.2 for more details). The median calculated $DF$ among the WWTPs in HydroWASTE is 570, and 2,533 (5.4%) of all plants showed a $DF$ value

below 10, i.e. lower than the recommended threshold for environmental regulations (EMA, 2006). Figure 5 shows the cumulative frequency distribution of $DF$s calculated from HydroWASTE.

As part of the methods to estimate missing attributes, Eq. (3) required the setting of a minimum $DF$ (see section 2.1.4 above) to estimate the upper limit of population served. We set this $DF$ value to be 5 and applied it to a total of 479 WWTPs, which represent 19% of all plants with $DF$s below 10.

### 370 3.3 Wastewater distribution in global rivers

To demonstrate the global utility of the HydroWASTE database, we here present a first application in which we used both the location of WWTPs outfalls and their associated attributes to route the discharged effluents along the global river network and calculate the ratio of wastewater in any river reach downstream of a WWTP in the database. The global assessment shows that more than 1.2 million kilometers of rivers are located downstream of WWTPs and thus contain some amount of WWTP

effluents (Table 6 and Fig. 6). Of these, about 96,000 km are located downstream of WWTPs that offer only primary treatment.



From all rivers containing wastewater, about one third (398,000 km) exceed a wastewater ratio of 1%. Over 72,000 km of impacted rivers surpass the wastewater ratio of 10% (i.e. corresponding to a dilution factor of 11), thus reaching or exceeding the recommended limit used in environmental regulations (EMA, 2006). Although 26% (19,000 km) of these highly impacted rivers are located within close vicinity of WWTPs (i.e., within the average distance of 8.5 km measured between the estimated

WWTP outfall location and the first river confluence thereafter) and may thus represent very local conditions and/or be affected by uncertainties in the WWTP locations, the remaining 74% (53,000 km) are further downstream from WWTPs, indicating persistent risks of high potential wastewater contamination. From the 15 countries with the highest total length of rivers containing any amount of wastewater, more than 10% of impacted rivers in China, Mexico, India, and South Africa exceed the 10% wastewater ratio in their discharge (Table 6). Finally, our study highlights several large river basins, including the

Hai (China), Mississippi (USA), and the Orange (South Africa) with particularly long sections of impacted rivers with wastewater ratios exceeding 10% (Table 7).

A total of 149,000 km of river stretches with a wastewater ratio exceeding 1%, and 31,000 km with a ratio exceeding 10%, are located along rivers that are currently considered to be free-flowing (Grill et al., 2019), i.e., rivers not substantially impacted by human activities that could alter their connectivity and ecosystem services. Furthermore, we estimate that 17% of rivers

that contain more than 10% of wastewater discharge are flowing through protected areas, defined as IUCN categories I-VI (UNEP-WCMC & IUCN, 2021). These results show that wastewater ratios could be used as an additional and complementary metric of water quality to be integrated in refined assessments of anthropogenic impacts on river health and ecological status. Finally, we assessed the number of potentially affected people along highly impacted rivers (i.e., rivers that carry at least 10% of wastewater). Following Richter et al. (2010), we assume that people living within 10 km of a river potentially dependent on

river services, such as water provision or groundwater recharge, or are exposed to risks related to river flows, such as flooding. With this definition, and using population information provided in the HydroATLAS database (Linke et al., 2019) we estimate that 874 million people live within 10 km of rivers with wastewater ratios exceeding 10%. As these people potentially use river waters for various purposes (e.g., drinking, cleaning, fishing, recreation), they are at elevated risk to be affected by water quality issues, including during floods.

**4 Data availability**

HydroWASTE including all described attributes can be accessed at https://doi.org/10.6084/m9.figshare.14847786 (Ehalt Macedo et al., 2021). *For review purposes the database can be accessed at this figshare link: https://figshare.com/s/27f064600a198d050403.*



## 5 Discussion and conclusion

Detailed water quality assessments require spatially explicit information on how, where, and how much wastewater is entering the river system. Here, we developed a global geospatial wastewater treatment plant database, HydroWASTE, involving the compilation of national and regional datasets, the georeferencing of all records to a river network, and the estimation of attributes not originally reported by the source datasets. HydroWASTE can be used for numerous applications ranging from environmental to human health risk assessments. It is the first database at the global scale that includes this level of detail and

comprehensiveness regarding geospatial WWTP locations, estimated effluent outfall locations, and associated attributes, such as population served, wastewater discharge, and level of treatment. In a first application, these characteristics allowed for the assessment of the distribution of wastewaters in the global river network.

Since WWTPs are important sources of contaminants into receiving waters, spatial information on wastewater discharge along with the key attributes are critical inputs to water quality modelling. The most recent global assessments did not have access

to this level of detail, relying on country-level statistics to account for these sources. The correct location of effluent discharge as a point source is rarely available, and if it is, it often does not connect with the river network integrated in the model. In this study we followed a conservative approach to topographically connect the point sources (WWTPs) with the river network. That is, instead of just connecting the WWTP to the nearest river reach, we introduced a tolerance of, on average, 6.5 km to allocate the outfall location further downstream, therefore connecting the WWTP to a river with larger expected discharge.

This intentional bias reduces the likelihood of incorrectly predicting low dilution factors and high contamination risks on small streams; however, this approach can also cause an underestimation of the true extent of affected rivers. Nonetheless, we consider this conservative approach to be particularly important given the uncertainties in the river network quality and the reported locations of WWTPs.

Even though our assessment does not consider any removal of contaminants caused by treatment, decay, abstractions or river

regulation, we argue that our results of wastewater ratios can serve as a first-order proxy to highlight areas of potential risk as persistent contaminants might not decay, could possibly (bio-)accumulate or be transported downstream all the way to the ocean.

As for dilution factors, we used long-term average natural river discharge for the calculations, thus we would expect higher concentrations of wastewater under low-flow conditions. We acknowledge that a high wastewater ratio in rivers will have

different implications in different regions, since treatment levels vary between countries and between individual WWTPs. In addition, this preliminary analysis does not account for any removal or decay processes along the river network. Nonetheless, our approach facilitates the identification of hotspots along rivers where wastewater ratios in river flows would be greatest and represent a risk to local ecosystem or human health. This information could be used to guide regional or even field studies to monitor or assess in more detail the actual local water quality.



## 5.1 Uncertainties

The uncertainties involved in this study mostly derive from the source datasets, which makes it difficult to trace their origins and calculate their effects on the final assessment. Some of the detectable inconsistencies relate to the reported attributes. For example, the coordinates do not always depict the precise location of the plant, but instead can refer to the location of the effluent outfall or an approximate location (note that each dataset is described in more detail in the Supplementary Information, section S1). To quantify this type of uncertainty, we verified the given locations for a reference subset of WWTPs which demonstrated the overall robustness of the applied approaches (see Supplementary Information, section S2.2).

HydroWASTE has extensive coverage of most European countries, the USA, India, and Canada, which represent the vast majority of WWTPs in the world (Table 4), and their records are based on information (location and most attributes) reported by their respective national datasets. For many of the remaining countries, especially those where the WWTP locations are sourced from the Open Street Map (OSM) web platform, their total population served tends to be underestimated in HydroWASTE as compared to country-level statistics, reflecting the incompleteness of WWTP records. An analysis between OSM and the available national datasets (see Supplementary Information, section S4.1) showed OSM to cover only 37% of the total number of reported facilities. In terms of estimating missing WWTP attributes, OSM-estimated wastewater discharge was compared to reported values from the South African national dataset, showing acceptable general agreement with 86% of the estimates ranging within one order of magnitude of reported values (see Supplementary Information, section S4.2). Overall, the lower-quality OSM-derived records constitute only 9% of the HydroWASTE database (representing 27% of population served and 19% of wastewater discharged).

As another source of uncertainty, the European WWTP dataset reports the population number as "population equivalent", which does not only refer to residents but also workers, tourists and service providers; that is, not only the country's permanent population with access to wastewater treatment, but the total ambient population using the sanitation services provided by the WWTPs. It can be argued that reporting in terms of "population equivalent" is more adequate when accounting for the amount and content of wastewater discharge (Daughton, 2012; Nakada et al., 2017); however, since some WWTPs also include industrial sources of wastewater, the number of people served can be overestimated (O'brien et al., 2014).

To indicate different levels of reliability for each attribute, including the WWTP location, several quality indicators were assigned to each record in HydroWASTE to help inform users about uncertainties inherent in the data. The quality indicators for population served, wastewater discharge, and level of treatment depend on whether the attribute is reported or estimated, and on the method used if estimated. The quality indicator for the WWTP location is based on a manual accuracy assessment performed using a global subset of the HydroWASTE database (see Supplementary Information, sections S2.1 and S2.2 for more details).

Despite these shortcomings, we believe that the 58,502 WWTPs in HydroWASTE and their effluent discharge into the environment provide a robust first-order representation of the majority of global domestic wastewaters.





## 5.2 Towards better representation of municipal wastewater discharge in the global river system

The robust and consistent global database presented here is designed to be used by water resource managers, policy makers, researchers, and public institutions to develop strategies to control, regulate or mitigate the impacts of anthropogenic
chemicals. It can be used to link population to individual WWTPs and trace the pathways of specific substances from households through certain treatment levels into the river network. In addition, HydroWASTE can be used to identify WWTPs for which an upgrade in technology would deliver the biggest improvement of downstream water quality. Alternatively, where necessary, the resulting predictions could identify where local regulations should be established to limit the release of problematic pollutants. And, finally, it is conceivable that this approach could be used to predict the potential impacts that
might occur with the development and anticipated widespread use of pharmaceuticals and household products, amongst other potential sources of contamination. Many applications of our novel database relate specifically to the Sustainable Development Goal (SDG) 6 ("Ensure access to water and sanitation for all") as it helps to provide reliable estimates of the distribution of wastewater to inform decision making that ultimately aims at achieving universal access of clean water globally.

In our efforts to obtain national datasets on WWTPs and their characteristics, we found that many countries (especially lower-
income ones) do not provide openly accessible information on these facilities in a consistent and comprehensive format. Given the many implications that WWTPs have on human and environmental health, either in their role to improve water quality through removing contaminants or as a potential point source of untreated substances, we strongly recommend that governments and international organizations produce and make publicly available the data that are required to support water quality assessments from local to global scales. In the interim, HydroWASTE can serve as a starting point for large-scale water
quality analyses, or as an initial framework to be expanded.

## Supplement

The supplement related to this article is available online at:

## Author Contribution

HEM compiled all datasets, estimated missing attributes, and performed the analyses. HEM, BL and JN developed the study
and drafted the paper. GG, JL, AL and RS contributed to the inclusion of national and regional datasets and their validation. All authors contributed to and approved the paper.

## Competing interests

The authors declare that they have no conflict of interest.



## Acknowledgements

We thank Edward R. Jones from Utrecht University for providing country-level estimates of wastewater information. This research was made possible through a grant from the Ideas Fund of the McGill Sustainability Systems Initiative (MSSI); the Natural Sciences and Engineering Research Council of Canada (NSERC Discovery grant RGPIN/04541-2019); and with the support of the James McGill Chair program of McGill University. We also thank various students in the research lab of BL who contributed to data entry and validation, including Dylan Marshall, Vicente Burchard-Levine, and Mathis Messager.

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





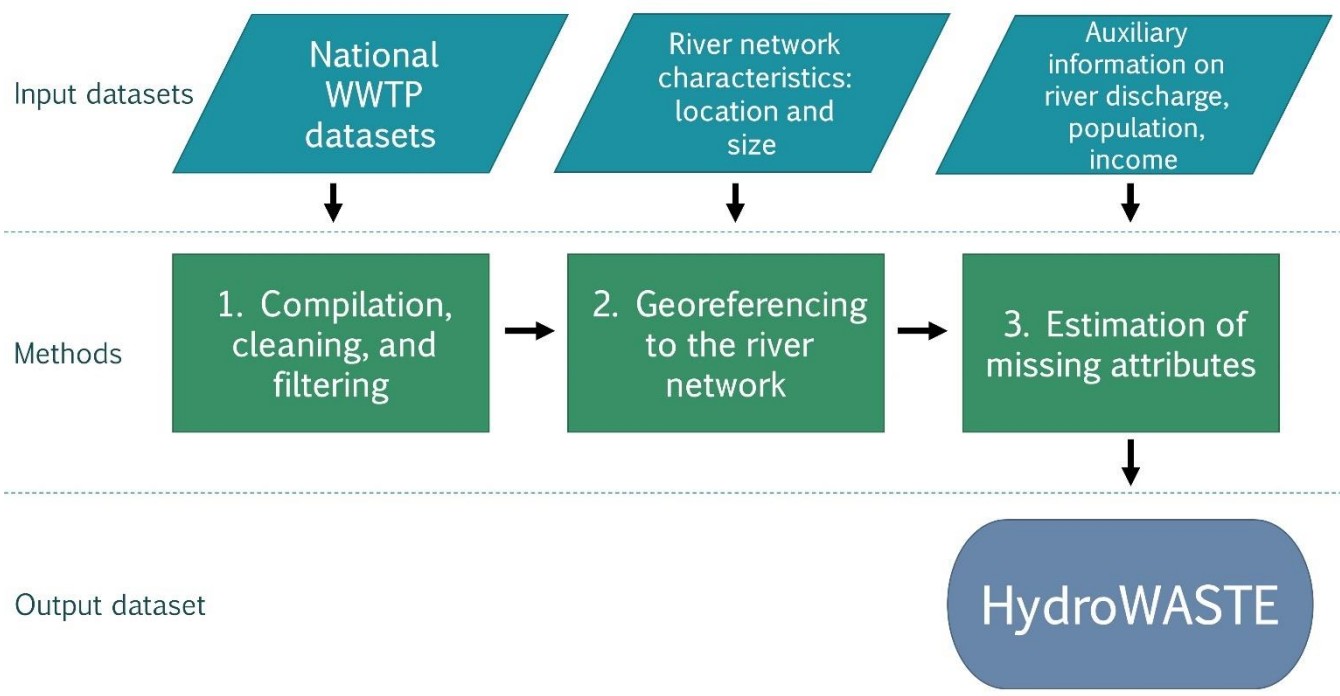


**Figure 1: Conceptual overview of the methodology used to create the global database of wastewater treatment plants HydroWASTE.**








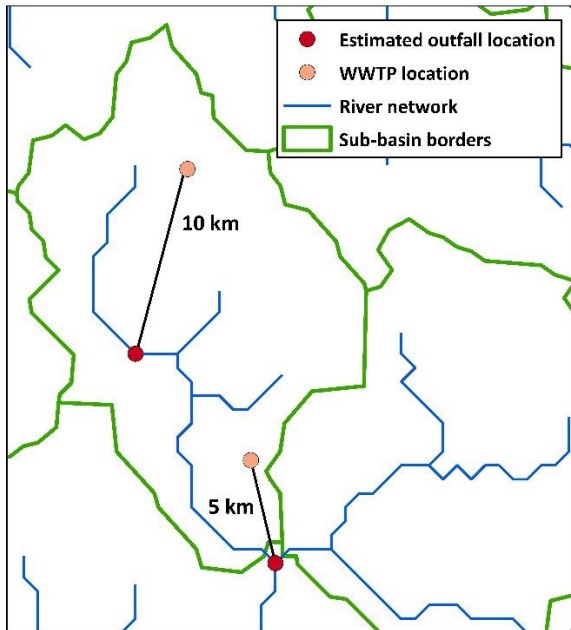

**Figure 2. Example of georeferencing process to assign WWTP effluent outfall locations. See text for more explanations.**


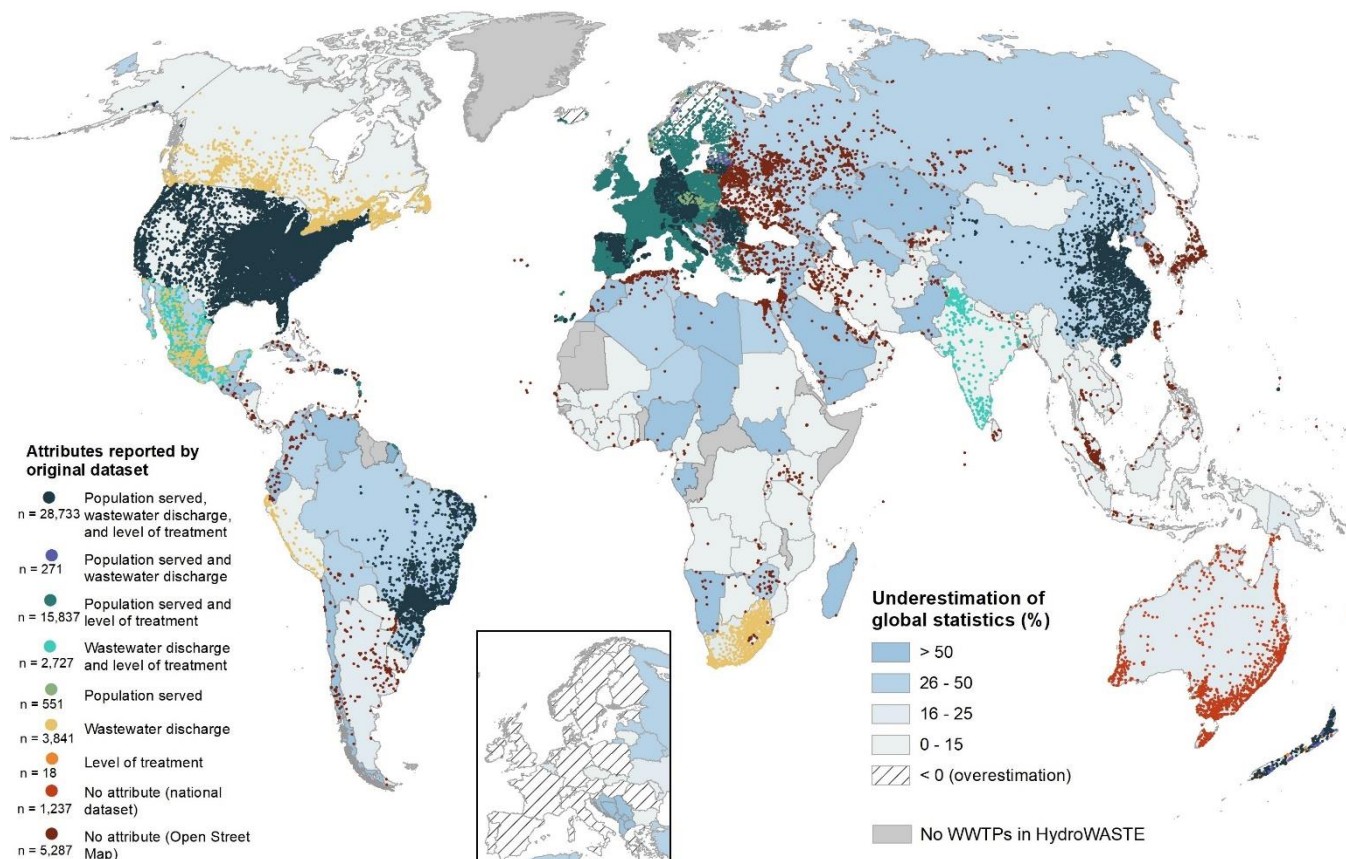

**Figure 3. WWTP locations, attributes, and completeness of 'population served' in HydroWASTE. Each point represents a WWTP, with colors depicting their reported attribute completeness with respect to population served, wastewater discharge and level of treatment. The country's area shading reflects the underestimation of the total population served per country in HydroWASTE as compared to global country statistics reported by JMP (WHO & UNICEF, 2017). Due to the high point density in Europe, an inset was added to show the underlying country shading.**




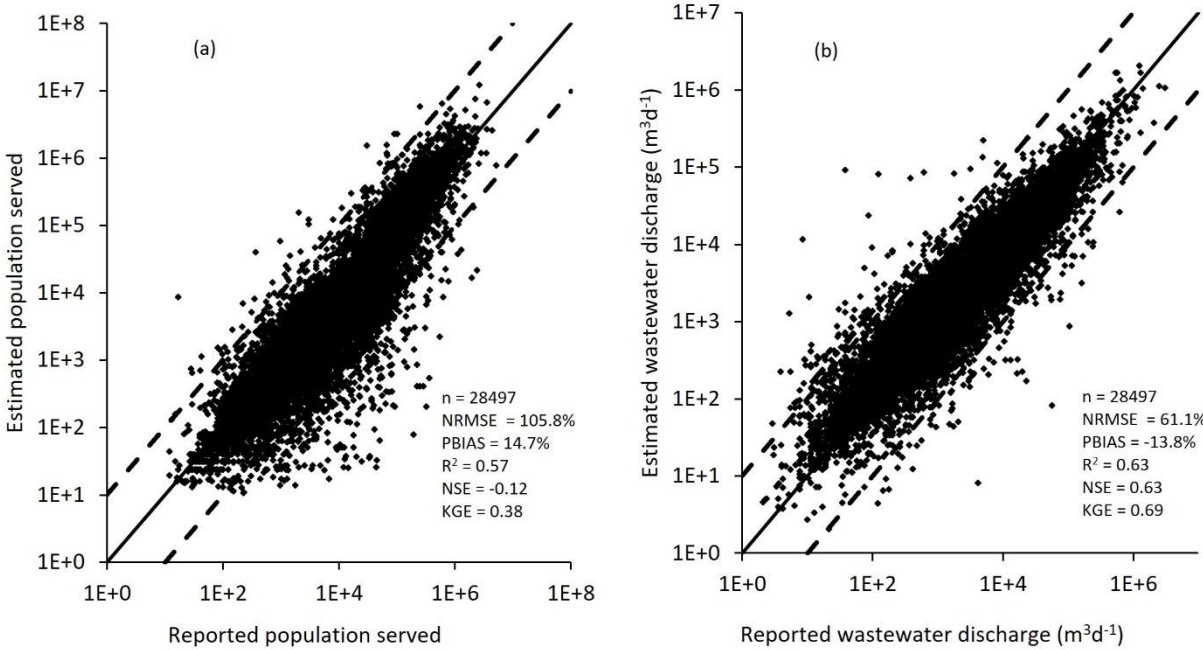

**Figure 4. Evaluation of the method used to estimate missing attributes (see text for more explanations): population served (a) and wastewater discharge (b). n is the number of records, NRMSE is the normalized root mean square error, PBIAS is the percent bias, NSE is the Nash-Sutcliffe efficiency and KGE is the Kling-Gupta efficiency. The solid line represents the 1:1 line and the dashed lines represents the error line of one order of magnitude.**




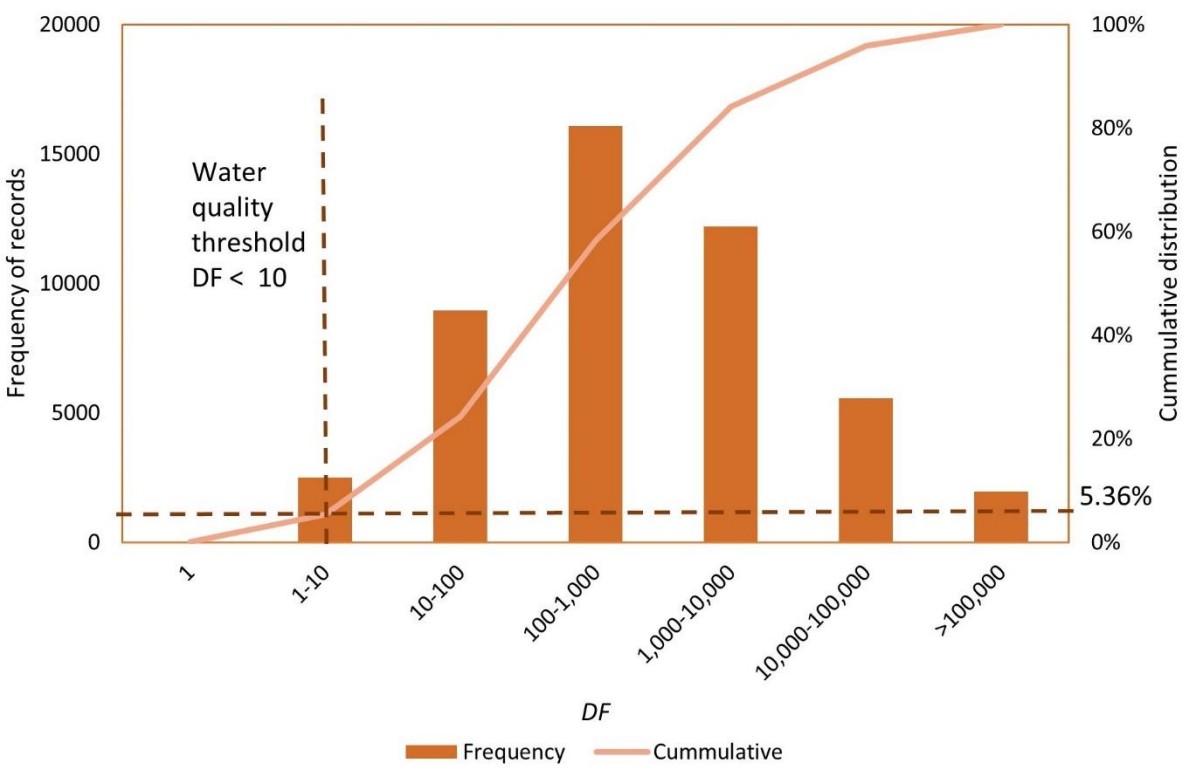

**Figure 5.** Frequency distribution of calculated dilution factors (DFs) of all WWTPs in the HydroWASTE database. WWTPs that are assumed to discharge directly into the ocean or a large lake are excluded from this analysis (see text for more information).







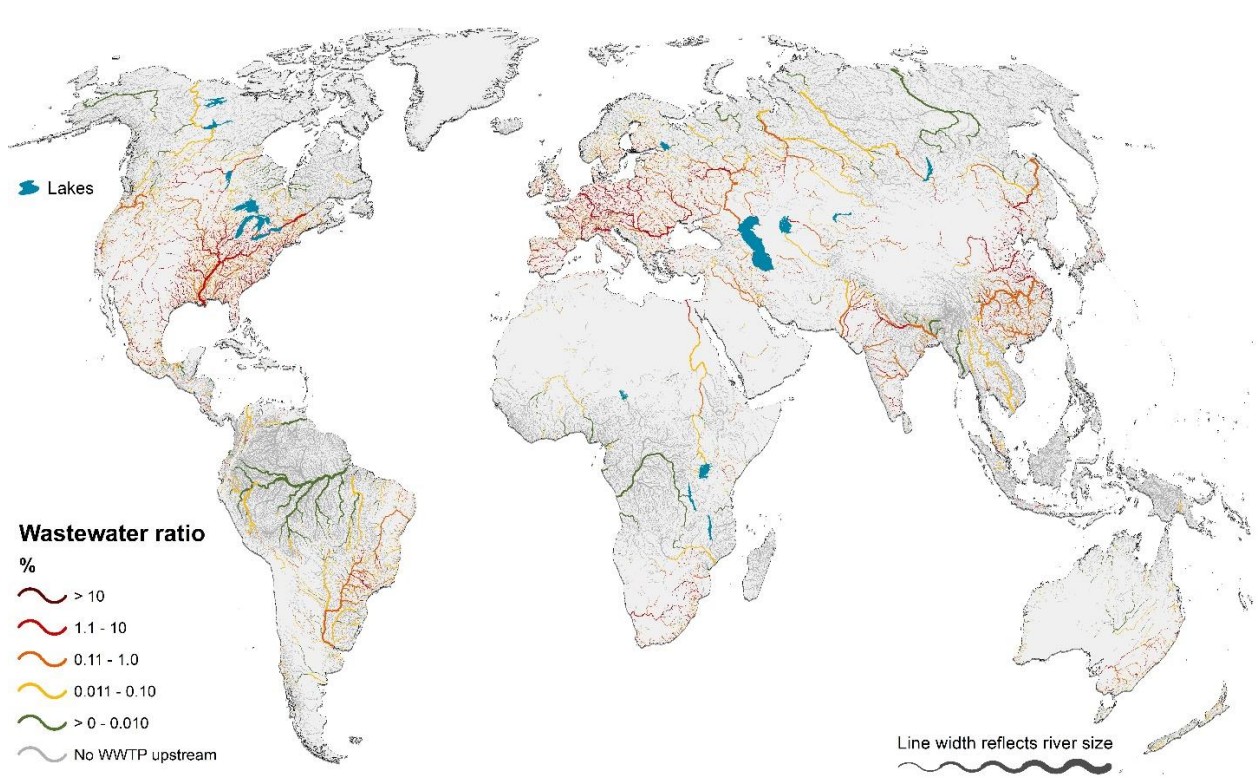


**Figure 6. Wastewater ratios in in the global river system.**



**Table 1. National and regional datasets included in the HydroWASTE database with their reported attributes. For comparison, country-level statistics of population served, as provided by the JMP-WASH database (WHO & UNICEF, 2017), and wastewater discharge, as provided by Jones et al. (2021), are listed. For more information on the individual datasets see Supplementary Information, section S1.**

| ID | Country/ region | Number of WWTPs included in HydroWASTE | Dataset name (source) | Population served (million) | | Wastewater discharge (million m³ day⁻¹) | | Treatment level | Year updated |
|---|---|---|---|---|---|---|---|---|---|
| | | | | Dataset | JMP-WASH | Dataset | Jones et al. 2021 | | |
| 1 | Europe | 24,971 | Urban Waste Water Treatment Directive (EEA, 2017) | 585.3 * | 451.9 | 55.2 | 101.1 | ✓ | 2020 |
| 2 | USA | 14,819 | Clean Watersheds Needs Survey (US EPA, 2016) | 258.1 | 262.9 | 126.8 | 125.4 | ✓ | 2012 |
| 3 | Brazil | 2,820 | Atlas Esgotos (ANA, 2017) | 71.7 | 129.9 | 11.3 | 20.7 | ✓ | 2017 |
| 4 | Mexico | 2,540 | Sistema Nacional de Informacion del Agua (CONAGUA, 2018) | ✗ | 94.9 | 11.9 | 9.0 | ✓ | 2018 |
| 5 | China | 2,486 | Ministry of Environmental Protection for China (Grill et al., 2018) | 480.8 | 810.1 | 93.9 | 85.5 | ✓ | 2010 |
| 6 | Canada | 2,064 | Wastewater Systems Effluent Regulations (WSER) (Environment Canada, 2017) | ✗ | 29.2 | 15.3 | 13.1 | ✗ | 2017 |
| 7 | Australia | 1,234 | National Wastewater Treatment Facilities Database (Hill et al., 2012) | ✗ | 21.1 | ✗ | 5.5 | ✗ | 2016 |
| 8 | South Africa | 964 | Department of Water Affairs | ✗ | 31.5 | 6.9 | 4.3 | ✗ | 2019 |
| 9 | India | 816 | The Central Pollution Control Board (CPCB, 2015) | ✗ | 132.1 | 23.3 | 9.8 | ✓ | 2015 |
| 10 | New Zealand | 317 | Wastewater Treatment Plant Inventory (Water New Zealand, 2019) | 3.5 | 3.8 | 1.3 | 0.9 | ✓ | 2019 |
| 11 | Peru | 184 | Plantas de Tratamiento de Agua Residual (SUNASS, 2018) | ✗ | 21.1 | 2.6 | 1.6 | ✗ | 2018 |
| 12 | Remaining Countries** | 5,287 | Open Street Map (OSM) | ✗ | 972 | ✗ | 137.4 | ✗ | 2020 |
| | Total | 58,502 | | 1,399.4 | 2,960.5 | 348.5 | 514.4 | | |

✓ Data mostly available with few exceptions

✗ Data not available

\* Capacity of WWTP reported as population-equivalent

\*\* The countries with the most records were Russia (1,269), Malaysia (484), Japan (378), Belarus (348), Turkey (319), and Argentina (143).







**Table 2: Summary of approaches used to estimate missing WWTP attributes based on auxiliary data.**

| Missing attribute | | Auxiliary data used to estimate missing attributes | Number of WWTPs without reported data (% of total) |
|---|---|---|---|
| Population served | with wastewater discharge available | Country-level treated wastewater discharge (Jones et al., 2021) Proportion of population using improved sanitation facilities (sewer connections) - JMP (WHO & UNICEF, 2017) | 6,568 (11.2) |
| | without wastewater discharge available | Proportion of population using improved sanitation facilities (sewer connections) - JMP (WHO & UNICEF, 2017) Population grid (Worldpop & CIESIN, 2018) | 6,542 (11.2) |
| Wastewater discharge | | Population served Country-level treated wastewater discharge (Jones et al., 2021) | 22,930 (38.1) |
| Level of treatment | | Population served GNI – World Bank (World Bank, 2019) | 11,187 (19.1) |






**Table 3. Top 20 countries that have the largest number of people served by WWTPs according to HydroWASTE database, and their total amount of wastewater discharge. The country-level statistics for population served and wastewater discharge were obtained from the JMP database (WHO & UNICEF, 2017) and Jones et al. (2021), respectively. Under/overestimation is calculated using the error percentage formula "((HydroWASTE - Country-level statistics) / Country-level statistics) \*100". For a complete list of all countries see Supplementary Information, Table S4.**

| Country | Number of WWTPs in HydroWASTE | Population served (million) | | | Wastewater discharge (million m³ day⁻¹) | | |
|---|---|---|---|---|---|---|---|
| | | HydroWASTE | JMP | Under /Over-estimation (%) | HydroWASTE | Jones et al. (2017) | Under /Over-estimation (%) |
| China | 2,486 | 481 | 810 | -41 | 94 | 86 | 10 |
| USA | 14,819 | 258 | 263 | -2 | 127 | 125 | 1 |
| India | 816 | 132 | 132 | 0 | 23 | 10 | 138 |
| Germany | 4,257 | 111 | 79 | 40 | 23 | 20 | 15 |
| Japan | 378 | 85 | 94 | -10 | 21 | 24 | -10 |
| Brazil | 2,820 | 72 | 130 | -45 | 11 | 21 | -45 |
| France | 3,622 | 72 | 54 | 32 | 13 | 10 | 32 |
| Italy | 4,090 | 70 | 56 | 24 | 15 | 11 | 36 |
| United Kingdom | 1,887 | 70 | 63 | 11 | 16 | 14 | 11 |
| Russia | 1,270 | 65 | 111 | -41 | 8 | 14 | -41 |
| Spain | 2,118 | 63 | 46 | 38 | 12 | 8 | 40 |
| Mexico | 2,540 | 58 | 95 | -39 | 12 | 10 | 32 |
| Egypt | 132 | 39 | 58 | -33 | 12 | 18 | -33 |
| Poland | 1,668 | 39 | 27 | 42 | 6 | 4 | 42 |
| South Korea | 87 | 37 | 49 | -25 | 14 | 19 | -25 |
| Turkey | 320 | 36 | 65 | -44 | 4 | 7 | -44 |
| Indonesia | 38 | 28 | 29 | -5 | 11 | 11 | -5 |
| Canada | 2,064 | 26 | 29 | -11 | 15 | 13 | 17 |
| South Africa | 964 | 25 | 31 | -20 | 7 | 4 | 62 |
| Colombia | 63 | 23 | 36 | -35 | 0.3 | 0.5 | -35 |
| **Total** | **46,439** | **1,790** | **2,257** | **-21** | **444** | **429** | **3** |
| **Global** | **58,502** | **2,297** | **2,960** | **-22** | **521** | **514** | **1** |






**Table 4. Global wastewater discharge and population served by WWTPs, according to HydroWASTE database and as provided by reported global values derived from country-level statistics. Reported WWTP data were provided by regional datasets (Table 1). Estimated WWTP data were derived using statistical methods (see section 2.1.4). n is the number of WWTP records in HydroWASTE.**

| Attribute | WWTP reported | | WWTP estimated | | Total in HydroWASTE | | Global values derived from country-level statistics | | |
| --- | --- | --- | --- | --- | --- | --- | --- | --- | --- |
| | Value | n | Value | n | Value | n | Value | Difference in HydroWASTE (%) | Source |
| **Population served (million)** | 1,399 | 45,392 | 898 | 13,110 | 2,297 | 58,502 | 2,960 | -22.4 | JMP Population with access to piped sewers (WHO & UNICEF, 2017) |
| **Wastewater discharge ($10^6$ m$^3$day$^{-1}$)** | 349 | 35,572 | 172 | 22,930 | 521 | 58,502 | 514 | 1.2 | Treated municipal wastewater (Jones et al., 2021) |







**Table 5. Level of treatment according to HydroWASTE database. The reported values are provided by the national datasets compiled and the estimated values were produced using methods described in section 2.1.4. The 'Correct prediction of reported treatment level' refers to the percentage of correct classifications using our prediction model.**

| Level of treatment | Number of WWTPs | | | Correct prediction of reported treatment level (%) |
|---|---|---|---|---|
| | Reported | Estimated | Total | |
| Primary | 765 | 116 | 881 | Not applicable* |
| Secondary | 25,681 | 8,960 | 34,641 | 73 |
| Advanced | 20,869 | 2,111 | 22,980 | 68 |
| Total | 47,315 | 11,187 | 58,502 | 70 |

\* No national dataset from a country in the low-income category was available that included this attribute.









**Table 6. Top 15 countries by total length of rivers downstream of WWTPs (for complete list of all countries see Supplementary Information, Table S5), and percentage of river length exceeding selected wastewater ratios.**

| Country | Total length of rivers downstream of WWTPs (km) | Percentage of river length downstream of WWTPs containing more than x% of wastewater | | | |
|---|---|---|---|---|---|
| | | x = 1% | x = 5% | x = 10% | x = 50% |
| United States | 287,395 | 33.5 | 9.6 | 5.0 | 0.6 |
| China | 104,698 | 52.8 | 27.9 | 20.2 | 4.1 |
| Brazil | 88,604 | 9.9 | 1.9 | 0.4 | 0.0 |
| Russia | 85,406 | 16.0 | 2.5 | 1.0 | 0.0 |
| Canada | 57,263 | 18.3 | 3.4 | 1.6 | 0.0 |
| Australia | 54,694 | 18.6 | 4.3 | 1.5 | 0.2 |
| Mexico | 43,657 | 37.9 | 18.4 | 11.5 | 1.8 |
| India | 33,425 | 53.7 | 26.2 | 18.4 | 4.2 |
| South Africa | 32,951 | 57.6 | 27.2 | 16.3 | 3.5 |
| France | 30,248 | 29.7 | 2.9 | 0.7 | 0.0 |
| Germany | 28,206 | 80.8 | 30.9 | 8.5 | 0.1 |
| Spain | 22,858 | 63.2 | 17.0 | 6.5 | 0.2 |
| Italy | 20,539 | 56.9 | 13.3 | 3.9 | 0.0 |
| Poland | 19,177 | 53.9 | 12.0 | 4.8 | 0.2 |
| Argentina | 17,933 | 6.9 | 1.6 | 0.9 | 0.0 |
| Total | 927,053 | 34.3 | 11.8 | 6.6 | 1.1 |
| Global | 1,214,362 | 32.8 | 10.9 | 5.9 | 0.9 |






**Table 7. Length of rivers containing more than 10% of wastewater in their natural discharge, by continent and for selected basins.**

| Continent | River basin | Length (km) |
|---|---|---|
| **Asia** | **Total** | **28,664** |
| | Hai | 6,346 |
| | Yangtse | 2,456 |
| | Huang He (Yellow) | 2,257 |
| | Ganges | 1,785 |
| **North America** | **Total** | **19,650** |
| | Mississippi | 5,499 |
| | Rio Grande | 2,641 |
| | Colorado | 2,466 |
| **Africa** | **Total** | **8,089** |
| | Orange | 2,904 |
| | Limpopo | 2,583 |
| **Europe** | **Total** | **13,191** |
| | Rhine | 1,881 |
| | Danube | 769 |
| **Oceania** | **Total** | **1,051** |
| | Murray | 348 |
| **South America** | **Total** | **1,593** |
| | Parana | 320 |
| **World** | **Total** | **72,237** |