# Peer review of "Distribution and characteristics of wastewater treatment plants within the global river network"

_Earth System Science Data, 2021_

## Referee Comment (RC2)

Review ESSD-2021-214 WWTP

High quality thoughtful effort to produce HydroWASTE.

Review link works, data download and open easily.

Do the authors correctly describe a database, or more properly a dataset. From looking at other recent examples in ESSD, this reviewer understands a database as open, accessible, recruiting (sometimes in easy automated fashion) additional incoming data (often from individuals as data providers), searchable through a variety of database fields, adopting and promoting novel community definitions, etc. Here a reader finds a static product (will the post-review version be on-line in interactive form?), with inputs entirely and solely at the discretion of the authors via 'official' national reports. Nothing in text about adding new data? Very good product, but probably more a dataset than a database? Produced with persistent skill but - even compared to HydroATLAS - this product seems more limited? Perhaps necessarily given the topic but authors provide no justification?

Lines 54, 55: exposure to chemicals from dense populations "which at a regional level can help prevent negative effects and determine hotspots of contamination." This in a discussion of importance of dilution factors but "prevent negative effects" requires higher dilution factors while "determine hotspots" inplies lower dilution factors. Perhaps authors mean 'or' rather than 'and'?

Line 126: "basis for calculating the contaminant loads" but authors already informed readers that for many so-called contaminants of emerging concern, WWTP accumulate but do not ameliorate contaminants of concern. Level of treatment irrelevant in that case? Authors could write here 'loads of treated or untreated contaminants'?

Line 335: "underestimations" of population served, of WWTP spatial density, of? A reader needs clarification here? In Fig 3, underestimation refers to population served, e.g. number of WWTP must be underestimated in view of population served? Where are the missing WWTP? Not addressed in uncertainty discussion, e.g. "reflecting the incompleteness of WWTP records" but no attribution noted or solutions proposed?k Authors tend to consider weaknesses in data sources (e.g. OSM) or in European methods for estimating populations served, but fundamental question of missing (or absent) WWTP not addressed? If comparison data products (e.g. Jones 2021, and note they describe their product as 'results' or as a 'study') come from same sources and therefore propagate same weaknesses, authors have no alternate forms of validation? Given variations among countries in terminology and reporting, basing comparisons on only South Africa seems very weak?

Line 363: here reader finds that "10,445" (20% of total WWTP in HydroWASTE) WWTP discharged into large (lake or ocean) bodies of water. But, back at line 326, readers learned that 224 (of 58,500 WWTP in HydroWASTE) discharged into oceans or lakes. Why this apparent discrepancy. If this reviewer missed the reason, other readers will likewise miss?

Line 706, Figure 6: Here authors show waste water fractions whereas throughout most of the manuscript they a focused on dilution ratios. If the latter represent the most-preferred regulatory factor, why not adopt them consistently. Or, if national reports differ, explain the difference? Not clear to this reader why we confront both DF and % WW fractions?

Line 752, Figure 3: Reference at top of column 7 should be Jones et al. 2021? Authors of this manuscript have taken global treated wastewater total from Jones et al. of 188.1 x 10xx9, converted to daily average to get 514? (I get 515?)

---

## Author Comment (AC2)

Response to Reviewer 1

**General comments**
**R1-C1**

*The study combines multiple existing wastewater treatment plant (WWTP) datasets, supplementing available information with additional (inferred) plant characteristics, to produce a consistent and coherent global dataset of wastewater treatment plants. Information captured within the dataset includes both geographical location and important plant characteristics such as flow rates and treatment level. The authors calculate an average wastewater treatment plant dilution factor, and link WWTP outfall locations to the river network, to calculate the length of the stream network downstream of WWTPs as a basic indicator for identifying areas of potential environmental concern. Overall, I believe this work to be both of high quality and interest for the scientific community. The manuscript clearly describes the dataset and is mostly well written, although I would recommend some edits to improve readability (particularly shortening sentences!). Therefore, after some revisions, I would recommend publication in ESSD. Please see my detailed comments below for my recommendations on how the manuscript could be further improved.*

We thank the reviewer for their very thoughtful comments on our manuscript and our database. They resulted in a more concise and transparent revised manuscript. We also apologize for not responding earlier to the comments, but we chose to wait for all reviews to be submitted before revising the entire manuscript to address all comments and suggestions raised. Any references to line numbers in our response relate to the original manuscript.

**Specific comments**
**R1-C2**

*I would revise the title to be more specific to the contents of the paper and dataset. "Their released effluents" is quite generic – for example this could suggest there is directly data on effluent quality (i.e. pollutant loads or concentrations) which is not contained within the dataset. Keep the focus on the spatial distribution and the core aspects of the dataset: volumes, populations served and treatment level.*

Thank you for this comment. We agree with your suggestion, but besides the WWTP characteristics, we also want to maintain the important notion of their linkage to the global river network. Therefore, we revised the title to: "*Distribution and characteristics of wastewater treatment plants within the global river network.*"

**R1-C3**

*It is stated in the abstract that WWTP information is particularly lacking in developing countries (which is true). This suggested that the dataset would improve this. However, the presented dataset is still vastly dominated by information from developed countries (particularly the USA and Europe with around 2/3 of the WWTP). Additionally, looking to the comparison to Jones et al., 2021; there is still a significant volume of wastewater treatment in the "remaining countries" that is not contained within the dataset. I appreciate the lack of data availability in many countries, however I think this should further be discussed in the manuscript.*

We understand the concern, but at the same time we believe HydroWASTE greatly improved the WWTP coverage in developing countries. The WWTP datasets from Brazil, China, and India, which we (for the first time) geolocated and included in a global database, cover large areas of developing countries, and as

they are based on official sources, we believe their overall coverage is adequate. The lower densities of WWTPs in Brazil, China, and India confirm that WWTP densities in the USA and Europe are indeed higher than elsewhere. For this reason, we believe that the distribution of WWTPs in reality is similar to the distribution as shown in HydroWASTE, even if we likely missed a higher percentage of facilities in some developing countries.

To reflect this idea in the manuscript, we added the following statement at line 346:

*"But even if we remove the datasets from USA, Europe, and Canada from the comparison, HydroWASTE still covers 92% of the reported treated wastewater discharge by Jones et al. (2021) (see Supplement, Table S5)."*

In addition, to address the lack of WWTP information from official sources of the "Remaining countries" at Table 1, we used point locations from Open Street Map (section 2.1.1) and inferred other characteristics using auxiliary datasets (including Jones et al., 2021). From the 138.3 million m3 day-1 estimated by Jones et al. (2021), we estimated 100.0 million m3 day-1 in HydroWASTE. We acknowledge and discussed the uncertainties involved in this process (5.1), especially between lines 444 and 452.

**R1-C4**

***"Wastewater discharge" is often used in the manuscript to refer to the outflow from wastewater treatment plants. But please note the term "wastewater discharge" could also be interpreted akin to "wastewater production" – the total amount of wastewater produced (regardless of if there is subsequent collection and treatment). Please consider renaming this throughout the manuscript (e.g. "discharge from WWTP" or "treated wastewater discharge") to make it clearer.***

We agree with the reviewer and changed "wastewater discharge" to "treated wastewater discharge" throughout the manuscript and database, where appropriate.

**R1-C5**

***Global wastewater in Table 1 vs. Table 3. Firstly, please make clear that this is treated wastewater discharge. Also, while the JMP-WASH and Jones et al., totals are consistent across both data tables, why are the "dataset" values different (e.g. global wastewater discharge in Table 1 = 5 million m3/day; while in Table 3 = 521 million m3/day)? Looking at the dataset, 521 million m3/day looks the correct value.***

As for the first comment, we rephrased "wastewater discharge" to "treated wastewater discharge", as suggested.

As for the second comment, our understanding is that the reviewer might have confused the column name "Dataset" (in Table 1), which indicates the information contained in each source dataset, with information contained in HydroWASTE (Table 3). To avoid this confusion, we renamed column 4 to "National/regional dataset (source)", and column 5 and 7 to "National/regional dataset".

**R1-C6**

***The authors do a very good job of inferring the outfall location (coordinates) of each WWTP, developing a rule based method in GIS. More as a comment, many (particularly global) water quality models are grid-based and typically at 30 arc-min (50 by 50km at the equator) or 5 arc-min (10 by***

*10km at the equator). So while the additional spatial resolution is good, it may be required to aggregated this back up to a gridded level for global water quality model runs, which may in turn cause a loss of usefulness of this data! Nevertheless, interesting data to provide and good that you also acknowledge these do not necessarily represent precise locations.*

We appreciate the positive comments of the reviewer regarding our work to geolocate the WWTPs. We also acknowledge the reviewer's concerns about the data being not immediately applicable at coarser resolutions. However, it was not a specific goal of our database development to create a database that is applicable in coarser, gridded models (we actually believe that the data by Jones et al. (2021) may be more appropriate for that). Upscaling our database would certainly be possible, but it is not a trivial task as spatial misalignments at the coarser resolution must be resolved. We thus decided to provide a product with the best spatial resolution possible and leave the decision to aggregate the database to a different level to each user and their individual requirements.

**R1-C7**

*For estimating the missing attributes for "population served"; I suggest the authors conduct and reflect upon the sensitivity of their results to their assumptions. The authors select the minimum value calculated from the three methods, but how do the results look if one method is used consistently instead? Also important to reflect upon the uncertainty of these derived attributes in general.*

The combination is used to reduce uncertainties of the individual methods, and we do agree that showing the sensitivity of results for each method could reinforce our decision to combine them. We therefore conducted an additional sensitivity analysis and added a section in the Supplement to present and discuss this analysis. To inform the reader about this addition, we added a reference in parenthesis in the main manuscript at line 285:

*"Finally, the minimum value among approaches A1, A2, or A3 was selected as the WWTPs estimate of population served (see Supplement, section S3.4 for an evaluation of each approach and the effect of using the minimum)."*

In the Supplement we then added the following section, starting at line 189:

*"S3.4 Evaluation of approaches A1, A2 and A3 to estimate missing records of population served*

*Figure 4a in the main manuscript shows the evaluation of the results of the entire method used to estimate population served. In this section we present separate evaluations of each approach (A1: estimating population served based on available treated wastewater discharge values; A2: estimating population served by summing the number of people living within a given radius of the WWTP; and A3: estimating the maximum population served based on dilution factors).*

*When applied to all existing validation data, approach A1 on its own (Fig. S2, left panel) shows a similar scatter plot and comparable goodness-of-fit parameters as the final method (Fig. 4a in main manuscript). This is because A1 provided the minimum (i.e., final) value more frequently than the other approaches in this evaluation (70%). Approach A2 (Fig. S2, right panel) resulted in more bias towards overestimation, especially for smaller WWTPs. However, even if approach A1 was generally superior to A2, approach A1 requires that a reported treated wastewater discharge value exists, which is not the case for 6,542 records of HydroWASTE.*

[Figure]

*Figure S2. Evaluation of the approaches A1 and A2 if used exclusively to estimate population served. n is the number of records, NRMSE is the normalized root mean square error, PBIAS is the percent bias, $R^2$ is the coefficient of determination, NSE is the Nash-Sutcliffe efficiency and KGE is the Kling-Gupta efficiency. The solid line represents the 1:1 correspondence line.*

*Approach A3 cannot be used exclusively to estimate population served, but only provides a maximum threshold of population served, which then can be applied to limit values calculated using approaches A1 or A2. Figure S3 shows how A3 prevents overestimation of population served as calculated by A1 and especially by A2.*

*Finally, in terms of applying the described estimation methods, the population served of 45,391 WWTP records are reported values as provided in the original data sources, and 13,111 records were estimated using the minimum value of approaches A1, A2, or A3. Approach A1 contributed a total of 4,125 estimated values, approach A2 contributed 8,425 estimated values, and approach A3 contributed the remaining 561 estimated values.*

[Figure]

*Figure S3. Evaluation of approach A3 (black dots) in comparison to A1 and A2 (red dots) for records where A3 presented the minimum value (n = 491)."*

**R1-C8**

*The manuscript consistently refers to the treatment levels as "primary", "secondary" and "advanced". While "primary" and "secondary" are very commonly described in the literature, the authors should confirm exactly what is meant by the description "advanced" (i.e. what specific treatment practices). More commonly used in literature is the term "tertiary" (describing a third level of treatment); with "advanced" or "quaternary" used to describe a fourth level of treatment (see work from van Puijenbroek1). Clarification here is very important for water quality modelers who need to define removal efficiencies of pollutants.*

We appreciate this comment and agree that readers might need a better explanation of the term 'advanced'. Therefore, line 120 now read as: *"...; and (5) the level of treatment offered by the WWTP classified as: primary, secondary, or advanced (which includes tertiary and any other processes that reduce the level of contaminants in the wastewater below that attainable through secondary treatment)*.

**R1-C9**

*Please acknowledge that the treatment capacities reported in the individual (source) datasets likely refer to maximum or operational design capacities, and thus might not actually represent volumes of wastewater being treated. For example, wastewater treatment plants may be shut or not operating at full capacity, leading to an overestimation of wastewater treatment2. This is very difficult to overcome, but should be acknowledged nevertheless.*

We generally agree with the reviewer's comment. However, we also observed that most datasets/records that we included in HydroWASTE actually provide both numbers, i.e., treated wastewaters and design capacity. For HydroWASTE, we always selected treated wastewaters if this choice was available. However, we agree that if this distinction was not available in the original data, the resulting uncertainty carries over to the final database. To better address this, we included new options at the column "QUAL_WASTE" in the database which now are: 1. Reported as 'treated' by regional dataset, 2. Reported as 'design capacity' by regional dataset, 3. Reported but type not identified, 4. Estimated.

The column "QUAL_POP" was similarly expanded to add more options to identify records that report population equivalent instead of population served: 1. Reported as 'population served' by regional dataset, 2. Reported as 'population equivalent' by regional dataset, 3. Estimated (with wastewater discharge available), 4. Estimated (without wastewater discharge available).

Finally, we added a sentence in the manuscript explaining our choices regarding treated wastewater discharge after line 148:

*Regarding the reported value of treated wastewater discharge, many national/regional datasets, including those of the USA and Europe, provided explicit values for both 'design capacity' and 'wastewater treated' (information on the availability of both attributes is noted in the HydroWASTE database for each country). Where available, we used 'wastewater treated' to refer to the amount of treated wastewater discharge."*

**R1-C10**

*It would be very interesting to see how the dilution ratio varies intra-annually, due to large seasonality in river discharge. However I do appreciate that the analysis presented in this paper is more with the*

*purpose of showcasing the HydroWASTE dataset. If possible, analysis of the seasonality in dilution ratios would further improve the manuscript, but is not essential.*

We agree with the reviewer that it would be very interesting to see the seasonality in dilution factors. However, it is not only the river discharge that varies intra-annually, but also the treated wastewater discharge, for which we do not have any temporal information. Still, following the reviewer suggestion, we decided to include an analysis of dilution factors and treated wastewater ratios under low flow conditions, using the same average treated wastewater discharges but instead of the long-term (1971-2000) average river discharge we used the minimum monthly discharge in an average year.

We added the following explanation introducing the minimum discharge in the "Data and methods" section, after line 168:

*"To assess dilution factors and treated wastewater ratios in the global river system at low flow conditions, we used the minimum discharge as provided in the HydroATLAS database, i.e., the lowest monthly flow value within an average year."*

We added and modified the sentences from line 364:

*"The dilution factors (DFs) were calculated for every WWTP record using Eq. (2), except for: (1) WWTPs that have their outfall location less than 10 km from large lakes or the ocean (n = 10,445), for which we assigned an infinite DF (see section 2.2 for more details); (2) WWTPs that reported wastewater discharge as 0 (n = 175); and (3) WWTPs not connected to the river network (n = 224). For average flow conditions, the median calculated DF among the analysed WWTPs in HydroWASTE (47,302) is 570, but 2,533 (5.4%) plants had a DF value below 10, i.e., lower than the recommended threshold for environmental regulations (EMA, 2006). For low flow conditions, the median DF decreases to 203 and the number of WWTPs having a value below 10 increases to 5,712 (12.1%). Figure 5 shows the cumulative frequency distribution of DFs calculated from HydroWASTE using average discharge (for low flow conditions see Fig. S6 in the Supplement)."*

Figure 5 now has a version for low flow conditions included in the Supplement.

[Figure]

*Figure S6. Frequency distribution of calculated dilution factors (DFs) at low flow conditions of all the WWTPs in the HydroWASTE database (for exceptions, see main manuscript, section 3.2.*

We also performed the same low flow analysis for the treated wastewater ratio. We added to line 384:

*"At low flow conditions, the length of rivers surpassing the wastewater ratio of 10% triples to over 213,000 km (17.6%). In addition, Germany and Spain join the list of countries with the largest percentages, both exceeding 30% of rivers that contain more than 10% of treated wastewater."*

Table 6 and Table S6 were also updated with the new information:

*Table 6. Top 15 countries by total length of rivers downstream of WWTPs, and percentage of river length exceeding selected treated wastewater ratios for average and low flow conditions (for a complete list of countries see Supplement, Table S6).*

| Country | Total length of rivers downstream of WWTPs (km) | Fraction of rivers downstream WWTPs containing more than x of treated wastewater (%) | | | | | | | |
|---|---|---|---|---|---|---|---|---|---|
| | | Average flow | | | | Low flow | | | |
| | | x = 1% | x = 5% | x = 10% | x = 50% | x = 1% | x = 5% | x = 10% | x = 50% |
| United States | 287,395 | 33.5 | 9.6 | 5.0 | 0.6 | 56.8 | 21.4 | 12.9 | 3.1 |
| China | 104,698 | 52.8 | 27.9 | 20.2 | 4.1 | 80.3 | 48.3 | 37.1 | 14.1 |
| Brazil | 88,604 | 9.9 | 1.9 | 0.4 | 0.0 | 23.9 | 7.6 | 3.7 | 0.3 |
| Russia | 85,406 | 16.0 | 2.5 | 1.0 | 0.0 | 37.9 | 13.8 | 5.4 | 0.3 |
| Australia | 57,263 | 18.3 | 3.4 | 1.6 | 0.0 | 50.9 | 26.4 | 19.9 | 14.1 |
| Canada | 54,694 | 18.6 | 4.3 | 1.5 | 0.2 | 24.5 | 6.4 | 3.7 | 0.3 |
| Mexico | 43,657 | 37.9 | 18.4 | 11.5 | 1.8 | 72.4 | 47.5 | 35.4 | 10.4 |
| India | 33,425 | 53.7 | 26.2 | 18.4 | 4.2 | 87.0 | 67.7 | 57.1 | 24.2 |

| | | | | | | | | |
|---|---|---|---|---|---|---|---|---|
| South Africa | 32,951 | 57.6 | 27.2 | 16.3 | 3.5 | 80.5 | 54.3 | 40.6 | 9.6 |
| France | 30,248 | 29.7 | 2.9 | 0.7 | 0.0 | 68.6 | 13.9 | 5.0 | 0.1 |
| Germany | 28,206 | 80.8 | 30.9 | 8.5 | 0.1 | 91.7 | 59.1 | 33.1 | 0.5 |
| Spain | 22,858 | 63.2 | 17.0 | 6.5 | 0.2 | 91.5 | 66.5 | 46.4 | 3.3 |
| Poland | 20,539 | 56.9 | 13.3 | 3.9 | 0.0 | 68.1 | 24.4 | 8.3 | 0.1 |
| Italy | 19,177 | 53.9 | 12.0 | 4.8 | 0.2 | 83.5 | 41.8 | 21.3 | 1.0 |
| Argentina | 17,933 | 6.9 | 1.6 | 0.9 | 0.0 | 18.4 | 10.4 | 7.3 | 4.0 |
| Total | 927,053 | 34.3 | 11.8 | 6.6 | 1.1 | 57.3 | 28.2 | 18.7 | 5.4 |
| Global | 1,214,362 | 32.8 | 10.9 | 5.9 | 0.9 | 55.9 | 27.1 | 17.6 | 5.1 |

Finally, we also included a version of Figure 6 for low flow conditions in the Supplement:

[Figure]

*Figure S7. Treated wastewater ratios in the global river system during low flow conditions*

**R1-C11**

*I find the analysis in 3.3 to be OK, but quite rudimentary. The focus is very much strongly on rivers (potentially) impacted by treated wastewater effluent as a result of the dilution factor, thus ignoring the treatment level. A 'low' dilution factor may not always be bad if the wastewater is adequately treated. Therefore, I would suggest to alter Table 7 to also include additional information, such as treatment level.*

We agree with the reviewer and revised the manuscript accordingly. Besides including the analysis of low flow conditions as described at R1-C10, we also added an analysis of the level of treatment. As a result, we included the following sentences starting at line 386:

*"However, a given wastewater ratio is expected to have different implications depending on the level of treatment offered by the WWTPs upstream. For example, although both the Mississippi Basin and Hai Basin have a comparable total length of rivers containing more than 10% treated wastewater, the higher percentage of advanced treatment in the Mississippi Basin may result in less environmental risk than the predominantly secondary treatment reported in the Hai Basin."*

Table 7 is now updated with information on the treatment level:

*Table 7. Length of rivers containing more than 10% of treated wastewater in their natural average discharge, by continent and for selected basins. The last three columns show the percentage of total treated wastewater discharged into rivers in each basin and continent by level of treatment (i.e., primary, secondary, or advanced).*

| Continent | River basin | Length of rivers containing more than 10% treated wastewater (km) | Percentage of total treated wastewater discharged into rivers by level of treatment | | |
|---|---|---|---|---|---|
| | | | Primary | Secondary | Advanced |
| Asia | Ganges | 1,785 | 7.6 | 92.4 | 0.0 |
| | Hai | 6,346 | 0.0 | 100.0 | 0.0 |
| | Yangtse | 2,456 | 0.0 | 100.0 | 0.0 |
| | Huang He (Yellow) | 2,257 | 0.0 | 100.0 | 0.0 |
| | **Total** | **28,664** | **2.2** | **84** | **13.8** |
| North America | Colorado | 2,466 | 0.0 | 21.7 | 78.3 |
| | Mississippi | 5,499 | 0.0 | 38.4 | 61.6 |
| | Rio Grande | 2,641 | 0.0 | 83.4 | 16.6 |
| | **Total** | **19,650** | **0.5** | **39.0** | **60.5** |
| Africa | Limpopo | 2,583 | 0.0 | 100.0 | 0.0 |
| | Orange | 2,904 | 0.0 | 100.0 | 0.0 |
| | **Total** | **8,089** | **0.6** | **99.4** | **0.0** |
| Europe | Danube | 769 | 1.0 | 26.2 | 72.8 |
| | Rhine | 1,881 | 0.0 | 5.2 | 94.8 |
| | **Total** | **13,191** | **0.2** | **34.8** | **65.0** |
| Oceania | Murray | 348 | 0.0 | 14.8 | 85.2 |
| | **Total** | **1,051** | **0.2** | **69.6** | **30.2** |
| South America and Central America | Parana | 320 | 2.9 | 88.3 | 8.9 |
| | **Total** | **1,593** | **1.9** | **75.5** | **22.6** |
| **World** | **Total** | **72,237** | **1.0** | **57.0** | **42.0** |

**Technical corrections**

**R1-C12**

***Line 9: Revise and shorten this sentence to improve readability.***

Agreed. Line 9 now read as: *"The main objective of wastewater treatment plants (WWTPs) is to remove pathogens, nutrients, organics, and other pollutants from wastewaters. After these contaminants are partially or fully removed through physical, biological and/or chemical processes, the treated effluents*

*are discharged into receiving waterbodies. However, since WWTPs cannot remove all contaminants, especially those of emerging concern,…".*

**R1-C13**

***Line 20: I would use "spatially-explicit" instead of "location-explicit".***

Agreed. The text has been revised as suggested.

***Line 38: Revise this sentence. E.g. "..reduce the pollutant loads reaching downstream water bodies, typically with a focus on organic matter and macro-pollutants"***

The text has been revised and now reads as: "*…to reduce the load of pollutants reaching downstream waterbodies, they typically focus on the removal of organic matter and macro-pollutants and not pollutants of emerging concern."*

**R1-C14**

***Line 44 - 48: Shorten this sentence.***

Line 44 now reads: "*Therefore, wastewaters are collected from municipal sources, transported to a location where they may or may not be treated, and then released into the environment. As a result, WWTPs serve as concentrated point sources of contamination to receiving waterbodies.*"

**R1-C15**

***Line 49: I would recommend citation of UNEP, 2016[3] and van Vliet et al., 2021[4] here.***

The suggested citations are now included.

**R1-C16**

***Line 55 – 58: Shorten this sentence.***

Line 55 now reads: "*However, to pinpoint which waterbodies are potentially affected by treated wastewaters discharged upstream, it is necessary to determine the location where these effluents are being released. This information can help in identifying which particular WWTPs should be targeted for the implementation of more stringent treatment standards and/or be upgraded through the deployment of advanced treatment technologies."*

**R1-C17**

***Line 108: "calculations are typically based…"***

The text has been revised as suggested.

**R1-C18**

*Line 246 and 285: There is unnecessary repetition here.*

We agree that there is repetition here. But it is intentional with the goal to increase clarity in the explanation of the multi-step methods. The first mention is part of the initial overview of the methods, the second mention is the explicit step within the calculations. We are afraid that if we removed one of the two instances, we might add confusion for the reader. So, we refrained from updating the text here.

**R1-C19**

*Line 327: The average distance between WWTP location and effluent outfall location seems quite large here? (I thought that this search radius was constrained to a maximum of 10km in most cases?). Please briefly elaborate on this.*

The average distance between WWTP location and effluent outfall location was found to be 6.5 km and thus relatively central within the range of 0 to 10 km that our search radius permits for most cases. We are not entirely sure why the reviewer considers this distance to be unexpectedly high. But maybe our explanation of the methods is somewhat confusing as we call the predefined radius a "maximum" search radius. Actually, most points were pushed towards 10 km, and even that is not a limit as explained at lines 216 to 221: *"If the closest location of connection to a river is further than 10 km, then the estimated outfall of the WWTP was georeferenced to that location, independent of distance, provided that all other rules still apply. In cases where the WWTP location is close to the sub-basin outlet, limiting the estimated outfall location to less than 10 km away from the WWTP location, the outfall location was additionally moved one grid cell (~500 m) further downstream; that is, into the next sub-basin and thus to a larger river, while keeping it close to the original WWTP location and in the same overarching basin (Fig. 2)."*

**R1-C20**

*Line 424 and 430: Some repetition here.*

Agreed. We updated the manuscript from line 424 to 434:

*"As for treated wastewater ratios, even though our assessment does not consider any removal of contaminants caused by treatment or decay processes in the river network, we believe the results can serve as a first-order proxy to highlight areas of potential risk to local ecosystems or human health. Persistent contaminants might not decay and could possibly accumulate or be transported downstream all the way to the ocean. Thus, our approach can facilitate the identification of hotspots along rivers where treated wastewater ratios would be greatest, and this information could be used to guide regional or field studies to monitor or assess the actual local water quality.*

*Nonetheless, it is important to acknowledge that a certain treated wastewater ratio in rivers will have different implications in different regions since treatment levels vary between countries and between individual WWTPs. In fact, the goal of this preliminary analysis is not to predict the actual distribution of contaminants, since WWTPs are not the only source of pollution. In 2020, 48% of the global population did not have access to wastewater treatment (WHO, 2021), thus forcing them to practise open defecation or to dump raw wastewater directly into waterbodies. The dimension of the global wastewater problem, including treated and untreated sources, goes beyond the scope of our analysis."*

**R1-C21**

***Figure 3: I appreciate the density of WWTPs in Europe and east-coast USA is very high and thus difficult to present visually, but perhaps slightly reduce the point size is possible..! Alternatively, you could perhaps consider only mapping wastewater treatment plants with a treatment capacity greater than a pre-defined threshold.***

We appreciate the reviewer's suggestions. We tried many versions of Figure 3, both for the original submission and now again for the revision. Our main goal of this figure is to carry as much information as possible and in particular to showcase the striking difference in the density of WWTPs in different regions. If we used a pre-defined threshold of treatment capacity, the figure would not reflect the difference in regional WWTP density. We also tried smaller point sizes, but this causes WWTP points in Africa or other low-density regions to be difficult to see. As we did not find a way to improve the figure without compromising other aspects, we left it as is and hope this is acceptable to the reviewer.

**R1-C22**

***Line 364: "but" instead of "and"***

The text has been revised as suggested.

---

## Author Comment (AC3)

Response to Reviewer 2

**R2-C1**

*High quality thoughtful effort to produce HydroWASTE. Review link works, data download and open easily.*

We thank the reviewer for their thoughtful comments on our manuscript and our database. The suggestions have helped to create a more concise and transparent manuscript. Please note that our comments to the other reviewers are now also available at the ESSD website. Any references to line numbers in our response relate to the original manuscript.

Please also note the supplement to this comment:

**R2-C2**

*Do the authors correctly describe a database, or more properly a dataset. From looking at other recent examples in ESSD, this reviewer understands a database as open, accessible, recruiting (sometimes in easy automated fashion) additional incoming data (often from individuals as data providers), searchable through a variety of database fields, adopting and promoting novel community definitions, etc. Here a reader finds a static product (will the post-review version be on-line in interactive form?), with inputs entirely and solely at the discretion of the authors via 'official' national reports. Nothing in text about adding new data? Very good product, but probably more a dataset than a database? Produced with persistent skill but - even compared to HydroATLAS - this product seems more limited? Perhaps necessarily given the topic but authors provide no justification?*

We understand that there are varying definitions of "dataset" and "database". In general terms (see e.g. https://www.usgs.gov/faqs/what-are-differences-between-data-a-dataset-and-a-database?), "*a database is an organized collection of data stored as multiple datasets.*" As HydroWASTE in its current version is stored as a single file, it could indeed be called a dataset. However, a dataset is also often considered to be a simple collection of data (in a table or spreadsheet) that is not intended to be updated, extended beyond its original intention, or manipulated by multiple users. HydroWASTE has been created by collating and curating multiple source datasets (i.e., it is a collection of different datasets, where several analyses were performed to include originally missing features). Also, it is designed to be expandable in the future, either through new data integration or inclusion of model results and relationships, and it has been enhanced by adding a linkage to the river network of HydroATLAS. We thus believe that our design and vision of HydroWASTE, starting with this version, fits more with that of a database than a dataset. However, we refrained from including more explanations on these intentions in the manuscript as we cannot ensure future updates at this point and do not want to raise false expectations. We hope that with these explanations the use of the term 'database' is acceptable in the case of HydroWASTE.

**R2-C3**

*Lines 54, 55: exposure to chemicals from dense populations "which at a regional level can help prevent negative effects and determine hotspots of contamination." This in a discussion of importance of dilution factors but "prevent negative effects" requires higher dilution factors while "determine hotspots" inplies lower dilution factors. Perhaps authors mean 'or' rather than 'and'?*

We understand the reviewers point. However, we find that one outcome is resulting from the other. For example, if we determine a hotspot using the model, we can prevent negative effects by avoiding further

contamination. To make this sentence clearer, we changed it to: "*..., which at a regional level can help prevent negative effects by identifying zones of high contaminant concentrations (i.e., "hotspots")*".

**R2-C4**

*Line 126: "basis for calculating the contaminant loads" but authors already informed readers that for many so-called contaminants of emerging concern, WWTP accumulate but do not ameliorate contaminants of concern. Level of treatment irrelevant in that case? Authors could write here 'loads of treated or untreated contaminants'?*

WWTPs are typically not designed to treat emerging contaminants, but different types and levels of treatment still might have an effect on any contaminant. But we agree that the reviewer's suggestion helps to avoid any confusion for the reader and we thus revised the text as suggested.

**R2-C5**

*Line 335: "underestimations" of population served, of WWTP spatial density, of? A reader needs clarification here?*

Thank you for identifying this ambiguity. We changed the text to specifically say "underestimations of population served".

**R2-C6**

*In Fig 3, underestimation refers to population served, e.g. number of WWTP must be underestimated in view of population served? Where are the missing WWTP? Not addressed in uncertainty discussion, e.g. "reflecting the incompleteness of WWTP records" but no attribution noted or solutions proposed?k Authors tend to consider weaknesses in data sources (e.g. OSM) or in European methods for estimating populations served, but fundamental question of missing (or absent) WWTP not addressed?*

We agree that this discussion is missing in the original manuscript. We thus added a new paragraph after line 452:

*"Besides the incompleteness of the OSM-sourced records, the national datasets may not include all facilities or may not have been updated recently. For example, the available datasets from the United States and China were last updated in 2012 and 2010, respectively, leaving around 10 years of new WWTP developments unaccounted for. This uncertainty could imply an underestimation of risk caused by missed WWTP effluents, and/or an overestimation of risk caused by an exaggeration of unserved populations in environmental assessments; although concurrent changes in total population numbers and/or treatment levels add to the complexity of recent developments."*

**R2-C7**

*If comparison data products (e.g. Jones 2021, and note they describe their product as 'results' or as a 'study') come from same sources and therefore propagate same weaknesses, authors have no alternate forms of validation? Given variations among countries in terminology and reporting, basing comparisons on only South Africa seems very weak?*

Unfortunately, we are not aware of any other validation data at the global level, and any national statistics that we know of typically are based on the same (or very similar) sources. So, the propagation of errors indeed remains a problem. However, a main goal of developing HydroWASTE was the linkage of

WWTPs to the river network (by using 'official' WWTP data sources wherever possible) rather than improving upon or quantifying the limitations of these 'official' data sources. Note that the mentioned analysis using South African data was only performed to evaluate the use of Open Street Map data as a valid source where no official sources were available. That being said, the OSM dataset was used for less than 10% of the HydroWASTE records.

**R2-C8**

*Line 363: here reader finds that "10,445" (20% of total WWTP in HydroWASTE) WWTP discharged into large (lake or ocean) bodies of water. But, back at line 326, readers learned that 224 (of 58,500 WWTP in HydroWASTE) discharged into oceans or lakes. Why this apparent discrepancy. If this reviewer missed the reason, other readers will likewise miss?*

As explained in original lines 326-327, the "224" WWTPs are located on small islands or in small coastal basins which do not have a river network, hence they are assumed to discharge directly into the ocean. In contrast, the "10,445" are connected to the river network, but were assumed to discharge into large lakes or the ocean for the calculation of dilution factors, due to their close vicinity to the lake or the ocean. To clarify this, we updated the sentence at original line 362:

"*The dilution factors (DFs) were calculated for every WWTP record using Eq. (2), except for: (1) WWTPs that have their outfall location less than 10 km from large lakes or the ocean (n = 10,445), for which we assigned an infinite DF (see section 2.2 for more details);...*"

Furthermore, we added the following explanation behind the related statement at line 314:

"*This conservative assumption was made to avoid the potentially erroneous assignment of very low DF values for WWTPs located near a large waterbody (but on a small stream) given the plausible option that the WWTP can discharge its effluents directly into the lake or ocean, e.g., by artificial over- or underground drainage, to increase dilution and ensure regulatory compliance.*"

**R2-C9**

*Line 706, Figure 6: Here authors show waste water fractions whereas throughout most of the manuscript they a focused on dilution ratios. If the latter represent the most-preferred regulatory factor, why not adopt them consistently. Or, if national reports differ, explain the difference? Not clear to this reader why we confront both DF and % WW fractions?*

The concepts of dilution factors and wastewater ratios are not the same and are not interchangeable. The dilution factor is used as a regulatory factor for WWTPs, i.e., it is determined for the WWTP itself by relating its effluent amount to the natural discharge right below the outfall location. In contrast, the wastewater ratios are calculated along the entire river system, based on the accumulated sum of upstream effluents, with 0% where there are no WWTPs upstream. We understand that Keller (2014) uses dilution factors as proxy for concentrations of contaminants, however the study works at a grid level, and does not consider the exact location of WWTPs, rendering the distinction of both metrics unnecessary.

Nonetheless, to avoid confusion we added/modified the following sentence at line 285:

"*Finally, since dilution factors are used only as a regulatory compliance factor for WWTP effluents, i.e., determined for each WWTP location individually, we also assessed the distribution of treated wastewaters throughout the entire global river network by calculating the ratio of accumulated upstream wastewater to natural discharge in every river reach.*"

**R2-C10**

*Line 752, Figure 3: Reference at top of column 7 should be Jones et al. 2021? Authors of this manuscript have taken global treated wastewater total from Jones et al. of 188.1 x 10xx9, converted to daily average to get 514? (I get 515?)*

The reviewer is correct about the citation and the total daily average, which is 515.3 million $m^3$ $day^{-1}$ instead of 514.4 million $m^3$ $day^{-1}$. Sorry for this error; it was caused by eliminating a few small (island) nations for which our own data (HydroWASTE) does not have any values. This error is now corrected, and the manuscript has been updated with the value 515. We also corrected the same error for the population served from the JMP. For this reason, Tables 1, 3, 4 and S5 were updated accordingly, as well as two locations in the manuscript where these numbers were referred to (original lines 343 and 345).

This correction did not affect the database or any conclusions made in the manuscript.

Reference:

Keller, V. D., Williams, R. J., Lofthouse, C., & Johnson, A. C. (2014). Worldwide estimation of river concentrations of any chemical originating from sewage-treatment plants using dilution factors. Environmental toxicology and chemistry, 33(2), 447-452.